# MPK3- and MPK6-mediated VLN3 phosphorylation regulates actin dynamics during stomatal immunity in *Arabidopsis*

Minxia Zou[1,6], Mengmeng Guo[1,6], Zhaoyang Zhou[2], Bingxiao Wang[1], Qing Pan[1], Jiajing Li[1], Jian-Min Zhou [3,4] & Jiejie Li [1,5✉]

Upon perception of pathogens, plants can rapidly close their stomata to restrict pathogen entry into internal tissue, leading to stomatal immunity as one aspect of innate immune responses. The actin cytoskeleton is required for plant defense against microbial invaders. However, the precise functions of host actin during plant immunity remain largely unknown. Here, we report that Arabidopsis villin3 (VLN3) is critical for plant resistance to bacteria by regulating stomatal immunity. Our in vitro and in vivo phosphorylation assays show that VLN3 is a physiological substrate of two pathogen-responsive mitogen-activated protein kinases, MPK3/6. Quantitative analyses of actin dynamics and genetic studies reveal that VLN3 phosphorylation by MPK3/6 modulates actin remodeling to activate stomatal defense in *Arabidopsis*.

[1] Beijing Key Laboratory of Gene Resource and Molecular Development, College of Life Science, Beijing Normal University, Beijing 100875, China. [2] Department of Vegetable Sciences, Beijing Key Laboratory of Growth and Developmental Regulation for Protected Vegetable Crops, China Agricultural University, Beijing 100193, China. [3] State Key Laboratory of Plant Genomics, Institute of Genetics and Developmental Biology, Innovation Academy for Seed Design, Chinese Academy of Sciences, Beijing 100101, China. [4] CAS Center for Excellence in Biotic Interactions, University of Chinese Academy of Sciences, Beijing 100049, China. [5] Key Laboratory of Cell Proliferation and Regulation of Ministry of Education, College of Life Science, Beijing Normal University, Beijing 100875, China. [6]These authors contributed equally: Minxia Zou, Mengmeng Guo. ✉email: Jiejieli@bnu.edu.cn

nfection of plants by folia pathogens involves pathogen penetration into inner tissues, where they obtain water and nutrients from internal cells. A variety of bacterial, oomycetes, and fungi exploit stomatal openings as major invasion routes. As a countermeasure, plants can rapidly close their stomata upon the perception of pathogens to restrict their entry. This control of stomatal closure, which is also known as stomatal immunity, is one of the first lines of plant innate immune responses[1,2]. Innate immunity is initiated by the recognition of microbe-associated molecular patterns (MAMPs) by cell surface receptors. Activation of receptors leads to pattern-triggered immunity (PTI), including activation of mitogen-associated and calcium-dependent protein kinases (MAPK and CDPK); bursts of cytosolic calcium and reactive oxygen species (ROS); production of defense hormones, such as salicylic acid (SA); and activation/inhibition of ion channels[3]. These signaling events ensure robust stomatal closure to prevent microbe invasion[1,2].

The actin cytoskeleton is a dynamic framework of cytoplasmic filaments that rearranges as the needs of the cell change during growth and development[4]. Increasing evidence points to the importance of the actin cytoskeleton for plant immunity[5]. A rapid increase in actin filament abundance occurs within minutes upon receptor activation in Arabidopsis epidermal cells, and this event is considered as a novel early hallmark of PTI responses. Many defense responses, such as ligand-induced endocytosis of receptors, organelle rearrangements, and targeted delivery of defense compounds to the infection site, are dependent on actin remodeling. Furthermore, cell wall fortification by callose deposition, apoplastic ROS production, and transcriptional reprogramming of defense genes is significantly impaired when the host actin cytoskeleton is disrupted. Thus, actin cytoskeleton and associated cellular processes are important for organizing intracellular and apoplastic defenses in host plants. When the actin cytoskeleton is perturbed, plants are more susceptible to both pathogenic and nonpathogenic microbes[5]. However, the precise functions of the actin cytoskeleton in host defense remain poorly understood.

In guard cells, the actin array undergoes dynamic reorganization, which is important for proper stomatal closure[6–13]. During stomata closure, the actin filaments reorganize from a radial array to a randomly organized network, and subsequently to a longitudinal alignment in the closed stomata. Disruption of this reorganization using genetical or pharmaceutical approaches all lead to impaired stomatal closure[7,14]. Li et al (2019) compared the dynamic behaviors of individual actin filaments between guard cells of stomata at the open and closed stage. They found that when stomata are open, radial actin filaments tend to be stabilized and bundled, whereas longitudinal filaments are more likely to be dissembled. On the contrary, longitudinal filaments exhibit reduced severing than radial actin filaments in the closed stomata[15]. Data from this study shed a light on how stomatal aperture-associated actin structures are maintained. It still remains unclear how one actin array arrangement would transit into another as a stoma closes. More importantly, what are the early dynamic events of actin filaments that possibly initiate the transition of the stomatal closure-associated actin structure remains unknown. To date, the majority of studies on actin organization in guard cells are performed under conditions of non-biotic stresses[7–10,12,13]. Shimono et al. (2016) examined the changes in actin architecture during MAMP- and pathogen-induced stomatal closure[11]. Their data suggest that actin array configurations in guard cells during immunity are similar to those during diurnal cycling[11,14]. However, details of actin dynamics during stomatal defense and underlying molecular mechanisms require further investigation.

Many actin-binding proteins (ABPs) can sense intracellular secondary messengers such as $Ca^{2+}$, phospholipids, ROS, and pH

and are modulated through posttranslational modifications (PTMs) such as phosphorylation or oxidation. ABPs are excellent candidates for transducing signals into cytoskeletal remodeling. Several ABPs have been shown to perceive early hallmarks of defense signaling and alter actin cytoskeletal dynamics to regulate defense[5]. Phosphatidic acid (PA) and ROS generated upon immunity activation inhibit the barbed-end capping activity of capping protein (CP)[16,17]. Porter et al (2012) suggest that actin-depolymerizing factor 4 (ADF4) is phosphorylated during plant defense response. This phosphorylation negatively regulates actin binding, and appears to be important for plant disease symptoms and the hypersensitive response phenotype[18]. A recent study reported that CPK3 phosphorylates ADF4 to regulate actin dynamics in response to MAMP and bacterial treatment. CPK3-mediated actin remodeling is important for plant resistance to bacterial infection[19]. In addition, quantitative phosphoproteomic analyses in Arabidopsis suggest that multiple cytoskeletal proteins are rapidly phosphorylated upon treatment with bacterial flagellin peptide mimic flg22. Several phospho-peptides from an Arabidopsis villin isovariant, villin3 (VLN3), have been reported[20,21]. A follow-up study further indicated that VLN3 is a potential MAPK substrate[21,22]. These studies suggest a role of VLN3 in plant innate immunity.

VLN3 belongs to the villin/gelsolin/fragmin superfamily; members of this superfamily are multifunctional proteins that are widely expressed in most eukaryotic cells[23]. The Arabidopsis genome encodes five isovariants of villin, VLN1 to VLN5. These villin isovariants are abundantly expressed in a wide range of tissues, with elevated expression levels in certain types of cells[24]. Except for VLN1, Arabidopsis villins exhibit actin filament bundling, $Ca^{2+}$-dependent severing, and barbed-end capping activities[25–28]. VLN3, together with its closest homolog VLN2, regulates actin bundle formation in cortical actin array, which is required for directional organ growth[28,29]. In this study, we uncover the mechanism by which VLN3-dependent actin rearrangements modulates PTI-induced stomatal closure. Moreover, VLN3 is phosphorylated by two key pathogen-responsive kinases, MPK3/6. Phosphorylation of VLN3 activates actin remodeling in guard cells to ensure robust stomatal immunity.

## Results

**Arabidopsis VLN3 plays an essential role in stomatal immunity.** Phosphoproteomic analyses show that VLN3 is phosphorylated upon MAMP activation[20]. This finding promoted us to hypothesize that VLN3 is required for plant immunity. To test this hypothesis, we inoculated homozygous loss-of-function vln3-1 and vln3-2 mutants with bacterial pathogen P. syringae pv. tomato DC3000 (DC3000). As shown in Fig. 1, when inoculated by hand infiltration of rosette leaves, both mutants showed enhanced susceptibility to DC3000 compared with wild-type (WT) plants. In experiments using spray-inoculation, the vln3 mutants supported significantly greater growth of bacteria (Fig. 1a, b). We also assessed the involvement of VLN2, the closest homolog of VLN3, in plant defense to pathogens (Supplementary Fig. 1a). The single vln2-1 mutant was not significantly different from WT when spray-inoculated with DC3000. Moreover, the vln2vln3 mutant did not show enhanced disease symptoms compared to the vln3 single mutant (Supplementary Fig. 1a), suggesting a dominant role of VLN3 in plant immunity. The enhanced bacterial growth on vln3 mutant by two types of inoculation methods suggest that both stomatal and basal resistance are affected by the loss of VLN3. In Arabidopsis, it has been shown that stomata close upon bacterial infection. This control of stomatal closure is known as stomatal immunity[30]. The restrictive role of stomatal closure on bacteria proliferation was

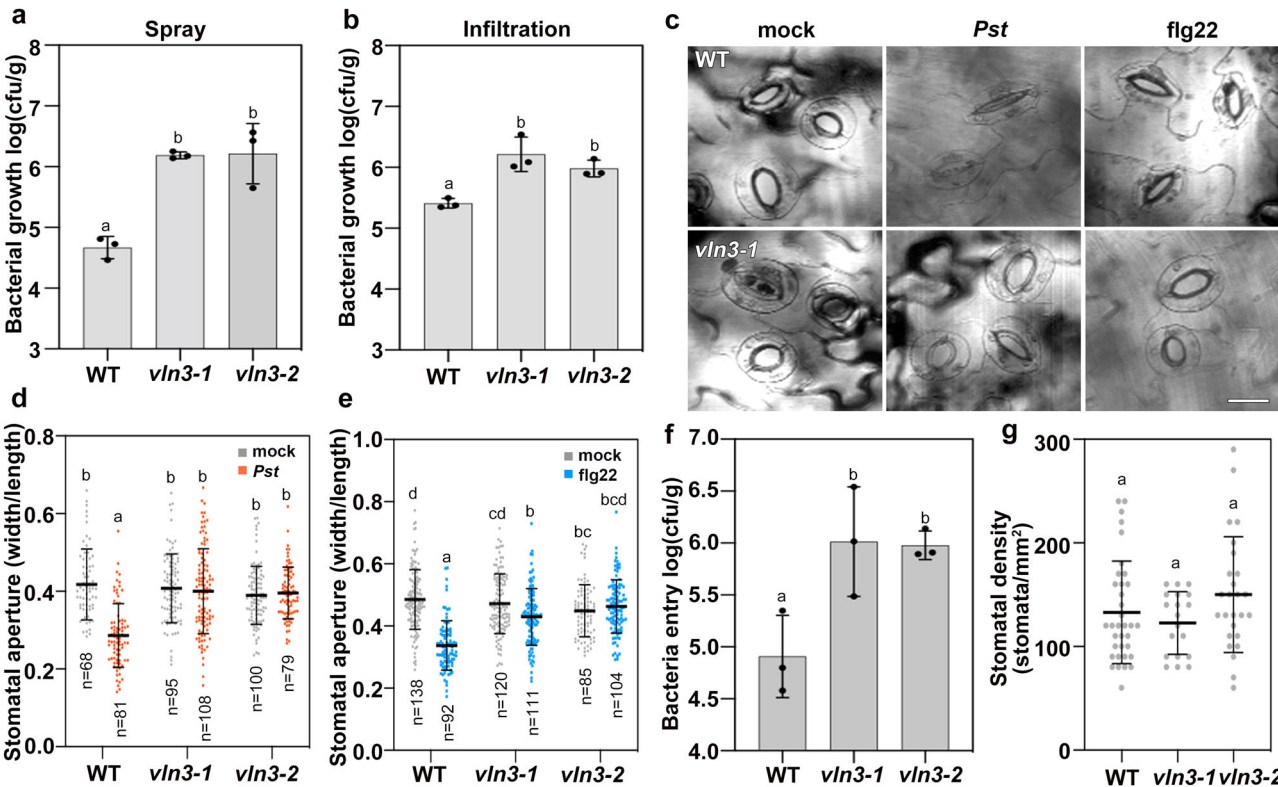

**Fig. 1 VLN3 plays an essential role in stomatal immunity in *Arabidopsis*. a–b** VLN3 mutants are more susceptible to bacterial pathogens. Plants of indicated genotypes were sprayed (**a**) or hand infiltrated (**b**) with *P. syringae* DC3000. Bacterial growth was measured at 2 d postinoculation (dpi). Representative images of stomatal closure in (**c**) or stomatal aperture measurements in (**d**, **e**) show impaired pathogen- or MAMP-induced stomatal defense in *vln3* mutants. Scale bars = 20 μm. **f** Loss of VLN3 leads to increased bacteria entry through open stomata. **g** Stomatal density was not altered in *vln3* mutants. Images for stomatal aperture analysis were also used to determine stomatal density. Value are means ± SD. $n = 3$ in (**a**, **b**, **f**). $n = 30$ images in (**g**). Sample sizes in (**d**, **e**) are indicated. Different letters indicate significant differences at $P < 0.05$, as determined by two-way ANOVA with Tukey's multiple comparisons test. The exact $p$ values are provided in the Source Data file.

more effective against bacteria inoculated onto the leaf surface compared with bacteria artificially infiltrated into the intercellular spaces[30]. Thus, the greater growth of DC3000 on *vln3* mutants infected by spray-inoculation (Fig. 1a), compared with hand infiltration (Fig. 1b), promoted us to study the role of VLN3 in stomatal defense.

The stomatal closure following treatments with DC3000 and flg22 were further examined. Consistent with previous study[30], both bacteria and MAMP trigger stomatal closure in WT plants, whereas stomata in *vln3* mutants were less responsive to these stimuli (Fig. 1c–e). Mutants of other villin isovariants (e.g., *vln2-1* and *vln4-1*) showed WT stomatal responses to MAMPs (Supplementary Fig. 1b). To investigate whether VLN3 is specifically involved in defense-associated stomatal closure, various treatments were applied. Abscisic acid (ABA), $CaCl_2$, and dark-induced stomatal closure were not significantly different between WT and *vln3*. Upon treatment with SA and $H_2O_2$, mutant stomata closed less than WT (Supplementary Fig. 1c). These observations indicate that VLN3 is required for optimal stomatal closure in response to bacterial signals, SA and $H_2O_2$. The importance of VLN3 in stomatal defense was further confirmed using a pathogen entry assay. This assay assesses the number of bacteria that enter into the apoplastic spaces of leaves in a given period of time[31]. We found that ~10-fold more bacteria entered into mutant leaf interior (Fig. 1f). This phenotype was not caused by the differences in stomata number because WT and mutant show similar stomatal density (Fig. 1g). VLN3 is broadly expressed including the expression in stomata[28,29] (Supplementary Fig. 1d). To further investigate whether VLN3 is involved in

PTI responses other than stomatal defense, we assessed several hallmarks of PTI in the *vln3* mutant, including callose deposition, MAPK activation, and ROS production. Neither flg22-dependent MAPK activation nor ROS production was altered by loss of VLN3 (Supplementary Fig. 1g, h). However, callose deposition was reduced in *vln3* mutants compared to WT (Supplementary Fig. 1e,f). Taken together, these data suggest that, in addition to stomatal defense, VLN3 is an essential component for mounting various immune responses in *Arabidopsis*.

**VLN3 is phosphorylated by MPK3/6 during innate immunity.** To verify VLN3 phosphorylation in vivo, full-length VLN3 was expressed in Arabidopsis protoplasts and total protein samples were subjected to the phos-tag gel analysis. No observable shift of VLN3 was induced by flg22 (Supplementary Fig. 2a). Because large phosphoproteins are often poorly resolved in phos-tag assays[32], we thus expressed truncated VLN3 fragments (N-VLN3: 1~719 amino acids, C-VLN3: 720~965 amino acids) in protoplasts. As shown in Fig. 2a, both unphosphorylated and phosphorylated VLN3 C-terminus were detected in mock-treated cells. Upon flg22 treatment, the lower C-VLN3 bands were shifted to a higher molecular weight, which was nearly abolished by the addition of phosphatase, whereas N-VLN3 bands were not upshifted after treatment with flg22 (Fig. 2a). These data suggest that the VLN3 C terminus is phosphorylated upon PTI activation. VLN3 has been suggested as a potential MAPK substrate[21,22]. In *Arabidopsis*, MPK3, MPK4, and MPK6 are rapidly activated upon plant perception of MAMPs. These kinases play critical roles in multiple plant defense responses[33]. We first examined the

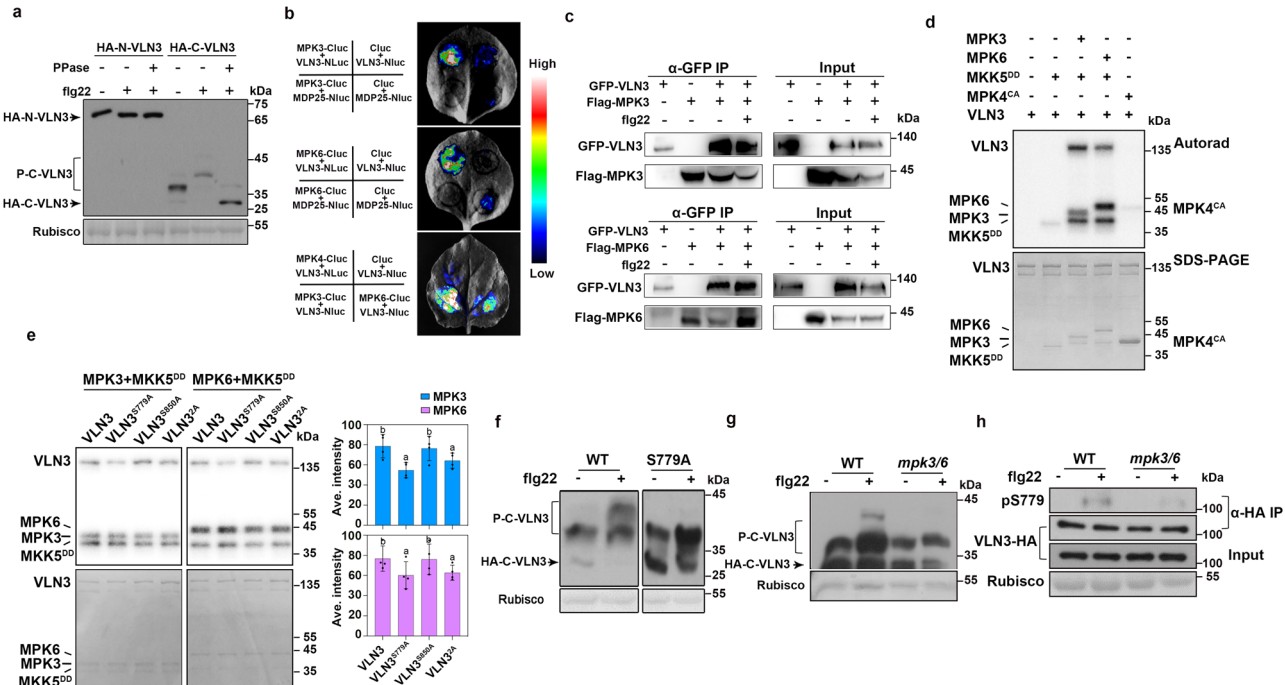

**Fig. 2 VLN3 is phosphorylated by MPK3/6 during plant innate immunity. a** The C terminus of VLN3 is phosphorylated upon flg22 activation. HA-tagged N-VLN3 and C-VLN3 were expressed in Arabidopsis protoplasts. Following treatment with mock or 100 nM flg22 for 10 min, total protein was separated in a phos-tag gel and VLN3 fragments were detected using anti-HA antibodies. Rubisco was used as a loading control. **b–c** MPK3/MPK6 interact with VLN3 in vivo. The indicated constructs were transiently expressed in *N. benthamiana*. Luciferase complementation imaging assay (**b**) and coIP assay (**c**) was performed. Nluc N-terminal fragment of firefly luciferase, Cluc C-terminal fragment of firefly luciferase. **d** Activated MPK3 and MPK6 phosphorylate VLN3 in vitro. VLN3 proteins were incubated with MPK3/6 activated by MKK5$^{DD}$ or a constitutively active form of MPK4 (MPK4$^{CA}$)[50] in an in vitro kinase assay. VLN3 phosphorylation was detected by autoradiography after gel electrophoresis. **e–f** Ser779 is required for VLN3 phosphorylation in vitro (**e**) and in vivo (**f**). **e** Phosphorylation of wild-type and mutated VLN3 was assessed using an in vitro kinase assay. VLN3 phosphorylation was detected by autoradiography after gel electrophoresis and the band intensity was quantified by densitometric analyses. **f** Protoplast expressing C-VLN3 and C-VLN3$^{S779A}$ tagged with HA were treated with flg22. Total protein was subject to phos-tag gel analyses. **g** MPK3/MPK6 are required for VLN3 phosphorylation in vivo. The HA-C-VLN3 was expressed in *MPK6SR* protoplasts. Prior to 10-min flg22 (100 nM) treatment, the *MPK6SR* protoplasts were treated with mock or NAPP1 (2.5 μM) for 12 h. The phosphorylation of C-VLN3 was detected by phos-tag gel analyses. **h** Flg22 induces Ser779 phosphorylation in a MPK3/6-dependent manner. HA-VLN3 was expressed in protoplasts of *MPK6SR* rosette leaves. The *MPK6SR* protoplasts were treated with mock or NAPP1 (2.5 μM) for 12 h before treatment with 100 nM flg22 for 10 min. HA-VLN3 protein was affinity purified with anti-HA antibodies, and Ser779 phosphorylation was detected by immunoblotting with anti-pSer779 antibodies. Total HA-VLN3 protein was detected by anti-HA immunoblot. Values in (**e**) are means ± SD. $n = 4$. Different letters indicate significant differences at $P < 0.05$, as determined by two-way ANOVA with Tukey's multiple comparisons test. The exact $p$ values are provided in the Source Data file. The experiments in (**a**, **c**, **d**, **f–h**) were repeated three times with similar results.

interaction between VLN3 and three kinases in vivo. Split-luciferase (split-LUC) complementation assays in *Nicotiana benthamiana* were performed (Fig. 2b). VLN3 was fused to the N terminus of luciferase (VLN3-Nluc), and MPK3/6/4 was fused to the C terminus of luciferase (MPK3/6/4-Cluc). The split-LUC assays showed that transient coexpression of VLN3-Nluc and MPK3/6-Cluc in *N. benthamiana* yielded strong luminescence signals, but no signal was detected when VLN3 and MPK4 were coexpressed. As a negative control, we failed to detect the interaction between MPK3/6 and another actin regulator MDP25 (Fig. 2b). Coimmunoprecipitation assays further confirmed the interaction between VLN3 and MPK3/6 (Fig. 2c). Total protein was isolated from *N. benthamiana* leaves expressing the construct *35 S:Flag-MPK3/MPK6* and *35 S:GFP-VLN3*. MPK3/6 was immunoprecipitated with anti-Flag antibody-conjugated agarose, and VLN3 was detected in the pull-down products by anti-GFP antibodies. Flg22 treatment did not obviously alter the interaction between the MPK3/6 and VLN3 (Fig. 2c). These data suggest that VLN3 interacts with MPK3/6 in vivo, and this interaction is independent of flg22 activation.

We next sought to determine whether MPK3/6/4 can directly phosphorylate VLN3. We purified recombinant VLN3 protein and performed in vitro kinase assays. As shown in Fig. 2d, recombinant MPK3 and MPK6 strongly phosphorylate VLN3. No phosphorylation signal was detected when VLN3 was incubated with MPK4 or MKK5$^{DD}$ (Fig. 2d). However, considering that the kinase activity of MPK4 is relatively low compared to activated MPK3/6, we cannot exclude a role of MPK4 in VLN3 phosphorylation. Given that MPK4 did not interact with VLN3 in the split-LUC assay, we then focused on MPK3/6. We next investigated which region of VLN3 is phosphorylated. As shown in Supplementary Fig. 2b, C-VLN3, but not N-VLN3 was phosphorylated by MPK3/6 (Supplementary Fig. 2b). The phosphorylated C terminus was subject to mass spectrum (MS) analyses to determine phosphorylation sites. Three phosphorylated serine residues were revealed (Ser779, Ser809, Ser850; Supplementary Fig. 2c). We mutated these residues to alanine (i.e., S779A, S809A, S850A), and assessed their requirement for phosphorylation using in vitro kinase assay (Fig. 2e). The Ser779 mutation significantly reduced VLN3

phosphorylation signals, supporting the notion that Ser779 is a key MPK3/6 phosphosite (Fig. 2e). Additionally, S779A largely reduced the motility shift of C-VLN3 induced by flg22 (Fig. 2f; Supplementary Fig. 3a), further confirming the importance of Ser779 for flg22-induced VLN3 phosphorylation. However, the Ser779 mutation neither completely abolish the VLN3 phosphorylation signals in vitro nor in vivo, indicating the contribution of additional phosphosites. To test whether VLN3 phosphorylation by MPK3/6 occurs in vivo, we expressed C-VLN3 in protoplasts from WT and the conditional loss-of-function mpk3/6 double mutants (MPK6SR, genotype: mpk3/6 pMPK6:MPK6$^{YG}$). The mpk3/6 loss-of-function mutant is lethal, but can be rescued with MPK6$^{YG}$, which is a 4-amino-1-tert-butyl-3-(1'-naphthyl)pyrazolo[3, 4-d] pyrimidine (NAPP1)-sensitized version of MPK6. MPK6$^{YG}$ loses its function after the addition of NAPP1[31]. As shown in Fig. 2g, the motility shift of C-VLN3 after flg22 treatment was abolished in mpk3/6 double mutants. We further developed an antibody recognizing phosphorylated VLN3 Ser779 (α-pS779). The pS779 antibody specifically detects VLN3 phosphorylation by MPK3/6, whereas the signal was absent in the controls with MPK4 or MKK5$^{DD}$ (Supplementary Fig. 3b). Anti-pS779 cannot detect the VLN3 S779A variant (Supplementary Fig. 3c). Immunoblotting with anti-pS779 revealed that VLN3 S779 was unphosphorylated in unstimulated WT but became strongly phosphorylated upon stimulation with flg22 (Fig. 2h), indicating that this residue is specifically phosphorylated in a flg22-dependent manner. In contrast, Ser779 phosphorylation was abolished in mpk3/6 double mutants (Fig. 2h), indicating that MPK3/6 are indeed required for the flg22-induced phosphorylation of VLN3 Ser779 in plant cells. Taken together, these results support that MPK3/6 phosphorylate VLN3 at Ser779 upon flg22 activation. To test whether VLN2 is also phosphorylated by MPK3/6, in vitro kinase assays were performed. However, we did not detect VLN2 phosphorylation by activated MPK3 or MPK6 in vitro (Supplementary Fig. 3d). Sequence alignment of Arabidopsis villins showed that Ser779 also exists in VLN4 and VLN5, but not in VLN1 and VLN2 (Supplementary Fig. 3e). However, VLN4 is not required for stomatal defense (Supplementary Fig. 1b). VLN5 is preferentially expressed in pollen tubes[26]. When comparing protein sequences between Arabidopsis VLN3 with villin-like proteins from rice, lily, and human, Ser779 is not conserved in plant and human villins (Supplementary Fig. 3f). These data indicate the specific requirement of villin proteins for different cellular processes.

**MPK3/6-mediated VLN3 phosphorylation is required for stomatal defense to bacterial infection.** To determine the biological significance of MPK3/6-mediated VLN3 phosphorylation, we substituted Ser779 with Ala, which blocks phosphorylation, or Asp, which often mimics phosphorylation, and introduced the mutant forms of VLN3 under the control of its native promoter in the vln3-1 mutant background. Flg22-induced stomatal closure was examined on these plants. As shown in Fig. 3a, the vln3 mutant did not close stomata after flg22 treatment, neither did the vln3 complementary lines carrying VLN3 with S779A mutation. However, the complementary lines carrying WT VLN3 or VLN3$^{S779D}$ transgene were fully responsive (Fig. 3a). The bacteria resistance was further analyzed. When spray-treated with DC3000, bacterial levels are similar between the WT and vln3 complemented with VLN3, or VLN3$^{S779D}$. However, vln3 plants complemented with VLN3$^{S779A}$ were still significantly more susceptible to bacterial infection (Fig. 3b, Supplementary Fig. 4a). Similar results were obtained when plants were inoculated by hand infiltration (Supplementary Fig. 4b). To investigate whether VLN3 phosphorylation is also involved in other PTI responses,

flg22-induced callose deposition was quantified. As shown in Supplementary Fig. 4c, VLN3$^{S779A}$ failed to restore the phenotype of vln3 mutants to the WT level, whereas VLN3$^{S779D}$ did (Supplementary Fig. 4c). Collectively, these results demonstrate that VLN3 phosphorylation is required for both stomatal and apoplastic defense.

We further asked whether phosphomimic VLN3 were able to recover the stomatal defense phenotype in mpk3/6 mutants. The WT VLN3, VLN3$^{S779A}$, and VLN3$^{S779D}$ were transformed into the MPK6SR plants. The bacteria resistance phenotype was assessed. In the absence of NAPP1, VLN3$^{S779D}$ led to an enhanced plant resistance to bacterial infection, whereas VLN3$^{S779A}$ failed to do so (Fig. 3c, d), confirming the positive role of VLN3 phosphorylation in plant defense. After treated with NAPP1, MPK6SR plants were more susceptible to spray-inoculated bacteria[31]. The VLN3$^{S779D}$ transgene reduced the bacterial population in MPK6SR plants, whereas the WT VLN3 and VLN3$^{S779A}$ did not (Fig. 3c, d). These results suggest that VLN3 phosphorylation plays an important role in MPK3/6-mediated plant immunity.

**VLN3 phosphorylated by MPK3/6 shows enhanced Ca$^{2+}$-dependent severing activity.** Actin reorganization is essential for stomatal closure[6–13]. We next asked whether and how MPK3/6-mediated VLN3 phosphorylation affect actin dynamics in stomatal defense. It was known that VLN3 severs actin filaments in a Ca$^{2+}$-dependent manner and bundles actin filaments[25]. We therefore investigated whether phosphorylation would affect these biochemical functions. The high-speed cosedimentation assay was performed to assess the actin filament-binding and -severing activity of VLN3. The low-speed cosedimentation assay was used to determine its bundling activity. In the high-speed cosedimentation assay, actin filaments sediment at 55,000 g. In the presence of an actin-binding protein that is capable of severing or depolymerizing filaments into small actin fragments or monomeric actin, there will be appreciable actin in the supernatant and less actin in the pellet. In the low-speed cosedimentation assay, however, actin filaments do not sediment at 13,500 g. Thus, actin-binding proteins that bundle or cross-link filaments into networks will lead to more actin in the pellet. As shown in Supplementary Fig. 5a, the amount of VLN3 cosedimented with actin was not affected by MPK3/6, suggesting that the filament-binding activity remains intact in the phosphorylated VLN3 (Supplementary Fig. 5a). When centrifuged at 13,500 g, the majority of actin was in the supernatant (Supplementary Fig. 5b). VLN3 addition caused more actin in the pellet, indicating VLN3-induced bundle formation. MPK3/6 did not exhibit a significant impact on the VLN3-induced actin sedimentation (Supplementary Fig. 5b). After incubation with Ca$^{2+}$, VLN3 shifted the actin from pellet to supernatant, suggesting the Ca$^{2+}$-dependent severing activity. This effect was even stronger when VLN3 was phosphorylated (Supplementary Fig. 5c). These data suggest that the phosphorylation modification enhances the filament-severing activity of VLN3, but has no obvious impacts on its ability to bind and bundle actin filaments.

To confirm the results above, we used total internal reflection fluorescence (TIRF) microscopy to visualize actin dynamics in real time. Prepolymerized Oregon-green-labeled actin filaments adhered to the cover glass of a perfusion chamber[25]. VLN3 proteins were perfused into the chamber, and time-lapse images were captured. Actin without VLN3 showed minimal breakage (Fig. 4a; Supplementary Fig. 6a; Supplementary Movie 1). The addition of VLN3-induced breaks along the filaments, demonstrating severing activity (Fig. 4b; Supplementary Fig. 6b; Supplementary Movie 2). Phosphorylated VLN3 generated more

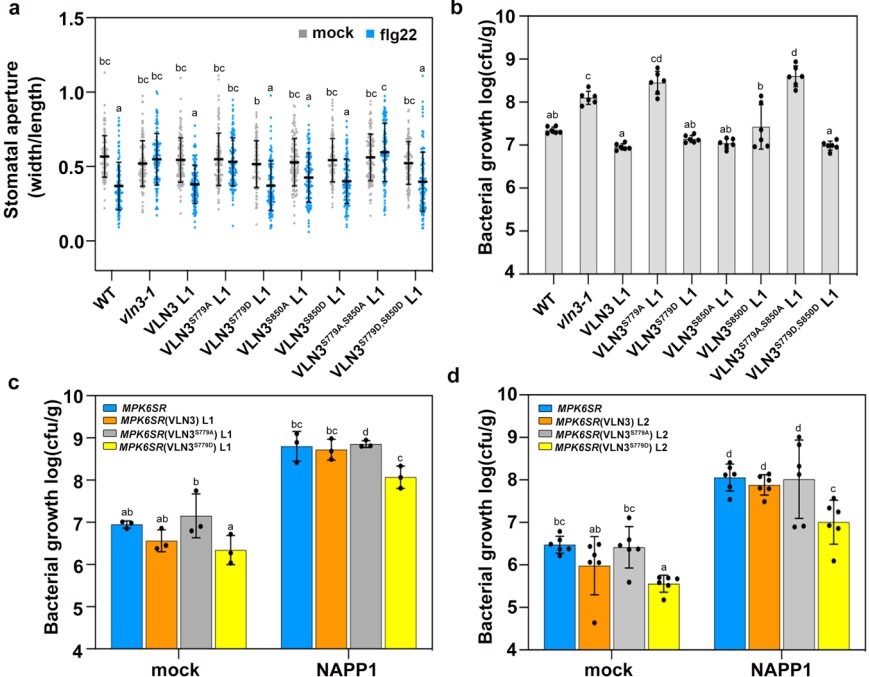

**Fig. 3 MPK3/6-mediated VLN3 phosphorylation contributes to stomatal defense.** The flg22-induced stomatal closure (**a**) and bacterial growth (**b**) was determined on the *vln3* T2 transgenic lines complemented with WT and mutant forms of VLN3. **c**, **d** Bacterial growth was determined in the *MPK6SR* overexpressing WT and mutant forms of VLN3. Data were obtained from two independent transgenic lines (L1 and L2). See also Supplementary Fig. 4a for bacterial growth data from the additional *vln3* complementary lines. Stomatal apertures were quantified after epidermal peels were treated with flg22 (10 μM) for 1 hr. The bacterial population in the leaf was determined 2 days after plants were spray-inoculated with DC3000. Plants in (**c**, **d**) were pretreated with mock or NAPP1 (10 μM) for 3 h prior to bacterial inoculation. Value are means ± SD. $n = 6$ in [**b**, **d**], $n = 3$ in [**c**] for bacterial growth measurements. $n = 110$ stomata from each treatment and genotype were analyzed for stomatal aperture; Different letters indicate significant differences at $P < 0.05$, as determined by two-way ANOVA with Tukey's multiple comparisons test in (**b**, **c**, **d**) or with Šídák's multiple comparisons test in (**a**). The exact $p$ values are provided in the Source Data file.

breaks along the filaments than nonphosphorylated VLN3 (Fig. 4c; Supplementary Fig. 6c; Supplementary Movie 3). The number of breaks per unit filament length per unit time (breaks $\mu m^{-1} s^{-1}$) was calculated as a quantitative measure for the severing frequency of each protein. VLN3 exhibited an average severing frequency of 0.0053 breaks $\mu m^{-1} s^{-1}$ at a concentration of 1 nM. The severing activity was markedly faster when the same amount of VLN3 was phosphorylated (Fig. 4g; Supplementary Fig. 6f). Actin filament length at a steady state was also examined. Prepolymerized actin was incubated with VLN3 in the presence of 10 μM free $Ca^{2+}$ for 15 min. When incubated without VLN3, actin filaments were on average 20 μm long (Fig. 4h; Supplementary Fig. 6g). In the presence of VLN3, however, the filaments were significantly shorter at ~7.8 μm in length (Fig. 4h; Supplementary Fig. 6g). VLN3 phosphorylation reduced filament length to ~3 μm (Fig. 4h; Supplementary Fig. 6g), which is consistent with enhanced filament severing.

We then hypothesized that Ser779 phosphorylation is responsible for the enhanced severing activity. In the case, we speculate that S779A mutation should abolish the function of phosphorylated VLN3, whereas VLN3$^{S779D}$ should be constitutively active. We first performed the high- and low-speed cosedimentation assay to test these hypotheses. S779A mutation did not affect the function of nonphosphorylated VLN3 (Supplementary Fig. 7a-c). However, VLN3$^{S779A}$ failed to increase filament severing as WT VLN3 did when activated MPK3/6 were added (Fig. 4d, g, h; Supplementary Fig. 6d, f, g; Supplementary Fig. 7d; Supplementary Movie 4), suggesting the requirement of Ser779 phosphorylation. We noticed that the severing was not enhanced by VLN3$^{S779D}$ in the absence of kinases(Supplementary Fig. 7c), this might be due

to the low sensitivity of this bulk actin assay. Thus, we performed the TIRF assay and observed that VLN3$^{S779D}$ exhibited an increased severing rate compared with nonphosphorylated VLN3, mimicking the function of phosphorylated VLN3 (Fig. 4e,g,h; Supplementary Movie 5). Taken together, these data further confirm the conclusion that MPK3/6-mediated phosphorylation at Ser779 enhances VLN3-induced actin filament severing.

**MPK3/6 and VLN3 are required for the actin array reorganization in both unstimulated and flg22-treated guard cells.** The data above showed that VLN3 phosphorylated by MPK3/6 increases actin turnover. To test whether this activity is required for stomatal closure-associated actin remodeling, we studied actin array reorganization during flg22-induced stomatal closure on *vln3*, *mpk3/6* double, and *vln3/mpk3/6* triple mutants. In unstimulated guard cells, the actin arrays in *vln3* guard cells were less bundled but more abundant than those in WT cells (Fig. 5a). The *mpk3/6* double and *vln3/mpk3/6* triple mutants exhibited a similar actin phenotype, but the defects were more severe (Fig. 5a). The actin differences between WT and mutants were further confirmed by skewness and density analyses, which are metrics used to estimate the extent of actin filament bundling and the percentage of occupancy of actin filaments, respectively[17] (Fig. 5b, c). These analyses demonstrate that MPK3/6 and VLN3 are involved in modulating actin organization in guard cells under normal conditions. A more dense and less bundled actin array is noted when MPK3/6 or VLN3 is absent in plants.

The cortical actin array undergoes rearrangement when guard cells were treated with flg22, and the actin filaments were radially

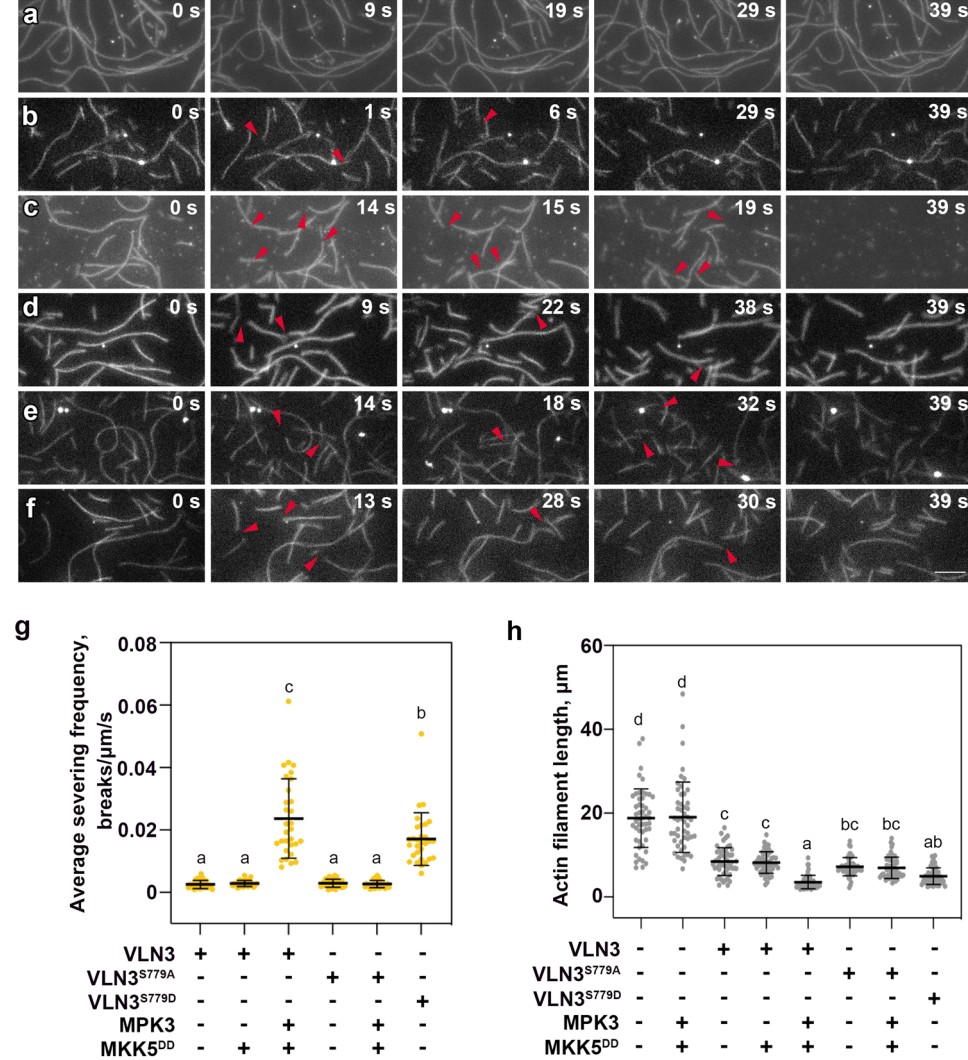

**Fig. 4 The phosphorylated VLN3 shows enhanced Ca$^{2+}$-dependent severing activity.** VLN3-mediated actin filament severing was visualized by time-lapse TIRF microscopy. Oregon-green labeled actin filaments adhered to the cover slip of a perfusion chamber, and then 1 nM wild-type or mutant VLN3 in the presence of 10 μM free Ca$^{2+}$ was perfused into the chamber. Time-lapse images were collected every second. Individual filaments showed breaks (arrows) along their length. The elapsed time in seconds is provided in the top right corner of each image (**a**, actin + MPK3 + MKK5$^{DD}$; **b**, actin + VLN3; **c**, actin + VLN3 + MPK3 + MKK5$^{DD}$; **d**, actin + VLN3$^{S779A}$ + MPK3 + MKK5$^{DD}$; **e,** actin + VLN3$^{S779D}$; **f**, actin + VLN3 + MKK5$^{DD}$). See also Supplementary Movies 1 to 6 online. Bars = 5 μm. **g** Severing frequency was calculated as the number of breaks per unit filament length per unit time. **h** Average actin filament length was measured for each indicated reaction. Value are means ± SD. $n = 30$ in (**g**), $n = 50$ in (**h**) filaments from at least 5 images for each treatment. Different letters indicate significant differences at $P < 0.05$, as determined by two-way ANOVA with Tukey's multiple comparisons test. The exact $p$ values are provided in the Source Data file.

oriented in open stomata at the beginning of the MAMP treatment. After prolonged flg22 treatment, a random organized actin array was observed. In closed stomata, the actin filaments were rearranged and bundled preferentially as long cables in the longitudinal direction (Fig. 5d). These actin rearrangements were similar to those observed during ABA-induced or diurnal stomatal closure[7,14]. In the absence of flg22, guard cells of WT and *vln3*, *mpk3/6* double, and *vln3/mpk3/6* triple mutants showed similar actin organizational patterns with radial actin dominating the cell population. Following flg22 treatment for 1 h, the proportion of WT guard cells with radial actin organization decreased, and actin filaments in the majority of guard cells became randomly or longitudinal distributed. By contrast, the actin networks in most of the *vln3* mutant guard cells stayed as radial and random organization; only approximately 20% of the guard cells exhibited a longitudinal actin organization. The actin defects in *mpk3/6* double or *vln3/mpk3/6* triple mutants were similar to those noted in

the *vln3* mutant (Fig. 5e). These organizational changes were further assessed by an analysis of actin filament angles in guard cells. As shown in Fig. 5f, an increase in the average filament angles occurred in flg22-treated WT stomata compared to their mock controls, indicating the occurrence of a more longitudinal actin array. Filament angles were slightly increased in the *vln3* mutant but not to a WT level after flg22 treatments. MAMP-induced increase in filament angles was abrogated in *mpk3/6* mutant. Importantly, there were no additive effects in the *vln3mpk3/6* triple mutants (Fig. 5f), suggesting that MPK3/6 and VLN3 function within the same immune signaling pathway. Collectively, these data suggest that both VLN3 and MPK3/6 are required for maintaining actin array organization in unstimulated guard cells. Moreover, the MPK3/6-VLN3 axis plays an essential role in actin array reorganization during flg22-induced stomatal closure.

It has been previously shown that an increase in actin filament abundance in epidermal pavement cells is an early hallmark of

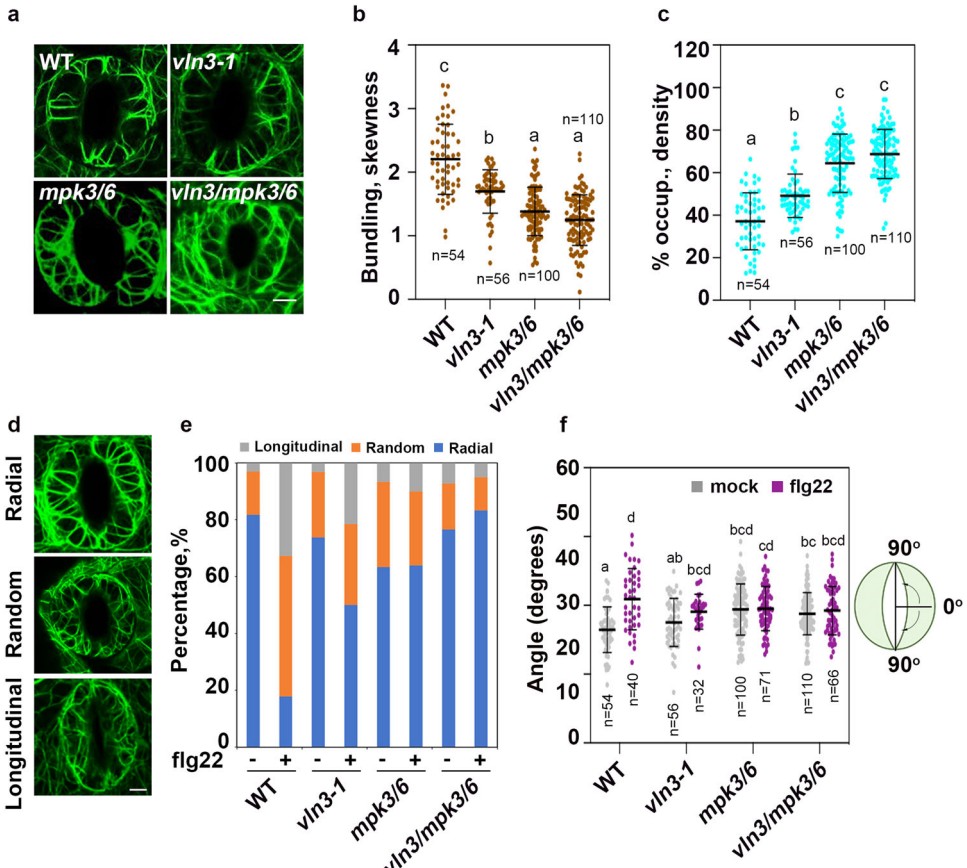

**Fig. 5 Actin reorganization during flg22-induced stomatal closure is impaired in *vln3*, *mpk3/6* double and *vln3/mpk3/6* triple mutants. a** Representative images of actin networks in wild-type and mutant guard cells treated with mock. Scale bars = 5 μm. **b** The extent of filament bundling or skewness was measured on images collected from guard cells of WT and mutant leaves. **c** Average filament density or percentage of occupancy analysis was performed on images used for (**b**). **d** Representative images of actin networks in wild-type guard cells during flg22-induced stomatal closure. Actin organization was classified into three groups: radial array, random meshwork, and longitudinal array. Scale bars = 5 μm. Percentage of these groups (**e**) and actin filaments angles relative to the width of stomatal pore (**f**) were calculated for indicated genotypes and treatments. Wild-type and *vln3* mutant leaves were treated with 10 μM flg22 for 1 h. Leaves of *MPK3SR* plants (genotype: *mpk3/6 pMPK3:MPK3^{TG}*)[31] were pretreated with 2.5 μM NAPP1 for 2 h prior to flg22 treatment. Value are means ± SD. The number of guard cells measured in (**b, c, f**) are indicated. Different letters indicate significant differences at $P < 0.05$, as determined by two-way ANOVA with Tukey's multiple comparisons test in (**c**), or with Šídák's multiple comparisons test in (**b, f**). The exact $p$ values are provided in the Source Data file.

PTI[5]. We also assessed the involvement of MPK3/6 and VLN3 in MAMP-induced actin responses in epidermal pavement cells. The actin density increases rapidly in WT treated with flg22, and the actin array in *vln3* or *mpk3/6* mutants showed a WT response, with increased actin density occurring within minutes (Supplementary Fig. 8). These data suggest that these proteins are not involved in MAMP-triggered actin remodeling in epidermal pavement cells.

**MPK3/6 and VLN3 regulate actin filament dynamics in mock- and flg22-treated guard cells.** The mechanism by which the radial actin array transitions into a longitudinal direction during stomatal closure remains poorly understood. It has been suggested that two stages are involved in this reorganization, with actin filament disassembly followed by filament reassembly when stomata close. Therefore, actin filament destabilization is a necessary aspect of this process[34,35]. Our results showed that MPK3/6-mediated VLN3 phosphorylation occurs within minutes upon flg22 treatment and the phosphorylation leads to enhanced severing in vitro (Fig. 4; Supplementary Fig. 6). Thus, we hypothesized that MPK3/6 and VLN3 might play a role in the stage of actin filament destabilization. To test this hypothesis, stomata were subject to a short-time

(10 min) flg22 treatment to investigate whether any early dynamic events occur that may initiate actin reorganization during stomatal closure. First, we tested whether MAMP treatment induces rapid changes in the overall dynamicity of cortical actin arrays. The correlation coefficient analysis provides a global view of changes in actin organization over time by calculating the correlation of the intensity of GFP-LifeAct signal at all pixel locations between all temporal pairs of images. The extent of actin rearrangements overtime was reflected by decay in the correlation coefficient as the temporal interval increased. The correlation coefficient values from actin arrays with reduced rearrangement exhibit less decay than a control population[36]. Using this method, we observed a rapid decrease in the overall actin dynamics within minutes after flg22 treatment (Fig. 6). Guard cells of *vln3* or *mpk3/6* mutants displayed a less dynamic actin array compared with WT cells when mock treated. The actin dynamicity in mutant cells was less sensitive to flg22 treatment compared to that in WT cells (Fig. 6). Collectively, these data suggest that actin arrays in guard cells respond almost instantly to immune signals. Additionally, MPK3/6 and VLN3 contribute to the overall actin dynamic response to flg22.

To gain further insights into the mechanism underlying the flg22-induced changes in actin dynamics, we studied the behaviors of actin at the single filament level and specifically

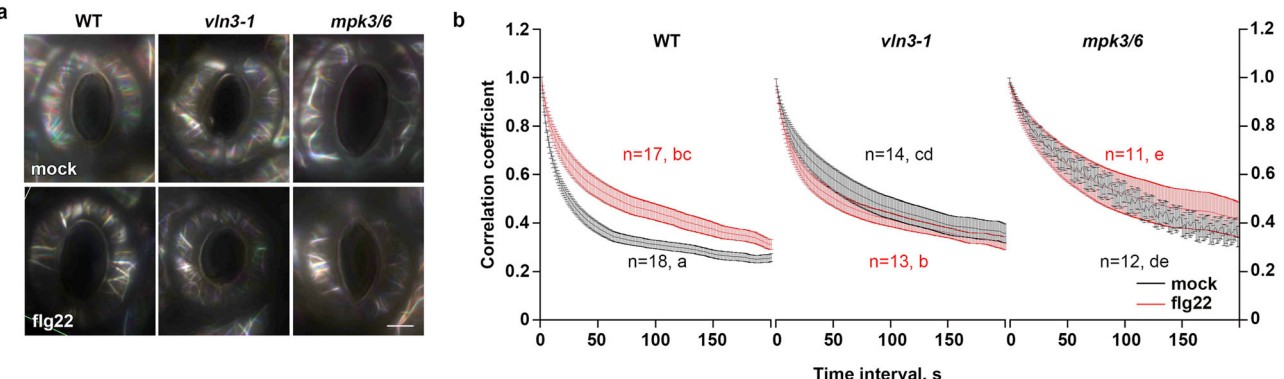

**Fig. 6 Dynamicity of the cortical actin array in WT, *vln3*, *mpk3/6* guard cells induced by MAMP. a** Actin array remodeling in guard cells is shown in indicated genotypes treated with mock or flg22. Images were merged from three images with 1-min intervals colored in red, green and blue. A white color indicates actin structures that remain relatively stationary during this time period. Scale bar, 5 µm. **b** Correlation coefficient analysis was performed on time-lapse series from WT and mutant cells treated with mock or 10 µM flg22 for 10 min. The extent of actin rearrangements or the overall dynamicity of the actin array was determined by decay in correlation as the temporal interval increased. Lower correlation values correspond with higher dynamicity of the actin array. Error bars represents SEM. Analyses were performed on time-lapse series taken from 10 leaves for each treatment and genotype. The sample sizes are indicated. Different letters indicate significant differences at $P < 0.05$, as determined by two-way ANOVA with Tukey's multiple comparisons test. Black or red letters indicate the data from mock or flg22 treatment, respectively. The exact $p$ values are provided in the Source Data file.

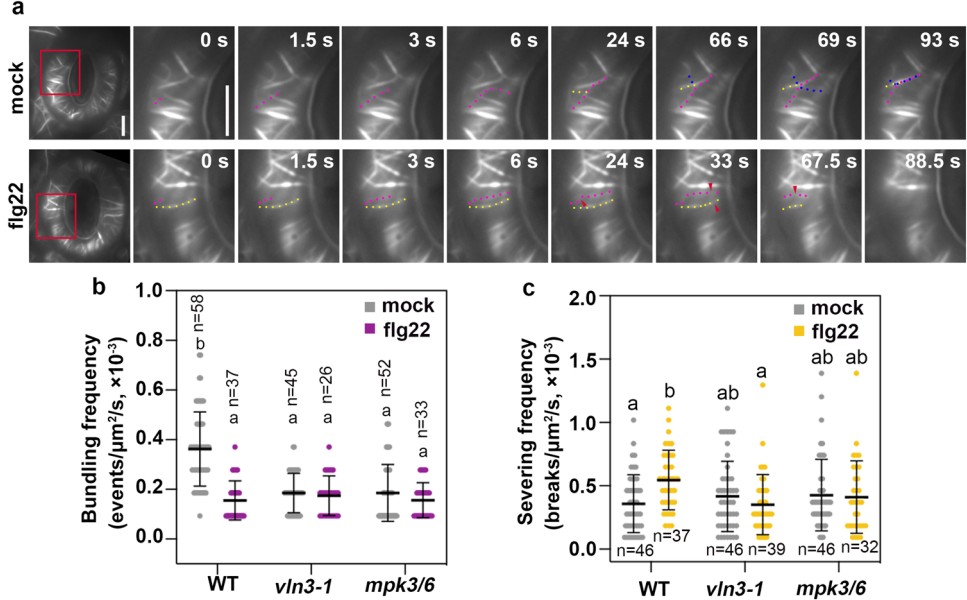

**Fig. 7 Flg22-induced actin dynamics in WT and mutant guard cells. a** Time-lapse images of single actin filaments in WT guard cells treated with mock or 10 µM flg22 for 10 min. In mock-treated cells, single actin filaments (red, yellow and blue dots) touch each other and are then bundle by a zippering mechanism (alternating dots). Representative actin filaments (red and yellow dots) from a flg22-treated cell were disassembled by numerous severing events (red arrows). See also Supplementary Movies 7 and 8 online. Scale bars = 5 µm. Actin filaments in WT cells treated with flg22 showed a significant reduction in bundling frequency (**b**) and increased severing frequency (**c**). However, flg22 treatments did not impact any of these parameters in *vln3-1* and *mpk3/6* mutant cells. Data are represented as means ± SD. Analyses were performed on more than 20 regions of interest from 20 time-lapse series taken from 10 leaves for each treatment and genotype. The exact sample sizes are indicated. Different letters indicate significant differences at $P < 0.05$, as determined by two-way ANOVA with Tukey's multiple comparisons test. The exact $p$ values are provided in the Source Data file.

focused on the parameters that are related to the biochemical function of VLN3. As shown in Fig. 7, the behaviors of actin filaments changed within minutes following flg22 treatment (Fig. 7). A twofold decrease in bundling frequency was noted (Fig. 7b). Moreover, the filament-severing frequency was significantly increased by MAMP treatment (Fig. 7c). It has been hypothesized that the stable open-aperture cable configuration acts as a cytoskeletal brake to stress-mediated guard cell closure. Moreover, the radial actin structure in open stomata needs to be removed or reorganized to ensure proper stomatal closure induced by external signals[8,10]. This hypothesis is supported by

our results. After MAMP treatment, the actin filaments were less frequently merged into bundles and quickly turned over. These dynamics behaviors may facilitate the disassembly of radial actin structure and initiate actin reorganization to promote flg22-induced stomatal closure. To investigate whether the MPK3/6-VLN3 axis contributes to these dynamic behaviors, we quantified these parameters in *vln3-1* and *mpk3/6* mutant cells following mock and flg22 treatment. In mock-treated cells, loss of VLN3 or MPK3/6 resulted in a significant reduction in bundling frequency, suggesting that actin filaments in the *vln3* single mutant and *mpk3/6* double mutant less frequently formed bundles. However,

these actin filaments exhibited a similar turnover rate to WT cells. These findings explain the more dense and less bundled cortical array in mutant cells. Importantly, the changes in filament behavior induced by MAMP were abrogated in the mutants (Fig. 7b, c), suggesting that MPK3/6 and VLN3 are responsible for the reduced frequency of filament bundling and increased filament turnover in flg22-treated guard cells. The actin dynamics in mock-treated *vln3* and *mpk3/6* mutant mimic the actin changes induced by flg22. However, stomata in mutants were not constitutively closed, indicating that changes in guard cell actin dynamics are necessary but not sufficient to trigger stomatal closure. Collectively, these data provide genetic evidence that MPK3/6 and VLN3 are required for flg22-induced actin remodeling in guard cells.

To determine whether the flg22-induced actin changes in guard cells requires VLN3 phosphorylation, we performed correlation coefficient analysis on transgenic plants carrying VLN3, VLN3$^{S779A}$, or VLN3$^{S779D}$. In unstimulated guard cells, wild-type and mutant VLN3 restored the defects in actin dynamicity of the *vln3* mutant. When treated with flg22, VLN3$^{S779A}$ failed to complement the actin response in the *vln3* mutant to the WT level, whereas VLN3 and VLN3$^{S779D}$ did (Fig. 8a,b). These data suggest that phosphorylation of VLN3 at Ser779 is required for flg22-induced actin dynamics during stomatal immunity. Of note, VLN3$^{S779D}$ expression did not elicit an actin response in the absence of flg22. Moreover, VLN3$^{S779D}$ plants did not show a constitutive stomatal defense (Fig. 3a),

suggesting S779 phosphorylation is required but not sufficient for flg22-induced actin dynamics and stomatal closure.

## Discussion

In this study, we demonstrate the role of the host actin cytoskeleton to limit bacterial infection by regulating stomatal immunity. Arabidopsis VLN3-dependent actin organization plays a positive role in this process. Loss of VLN3 fails to activate stomatal immunity to prevent bacterial entry into plants, leading to enhanced plant susceptibility to bacterial infection. Upon flg22 stimulation, VLN3 is rapidly phosphorylated by defense-responsive kinases, MPK3, and MPK6. Ser779 is required for VLN3 phosphorylation. This phosphorylation modification significantly enhances the severing activity of VLN3 but has no apparent impacts on other biochemical properties of this protein. Loss of VLN3 or MPK3/6 results in similar actin defects in guard cells. The actin filaments in the *vln3* single mutant and the *mpk3/6* double mutant merge into bundles less frequently than those in WT cells, resulting in a more abundant and less bundled actin array. Additionally, we showed that flg22 induces significant changes in the dynamic behavior of actin filaments in guard cells. MAMP treatment destabilizes actin filaments by decreasing the frequency of bundle formation and increasing the turnover rate. These features may lead to the disruption of the radial actin array in the open stomata, which facilitates actin reorganization to initiate stomatal closure. These dynamic parameters are insensitive to flg22 in the *vln3* and *mpk3/6* mutant, suggesting that the

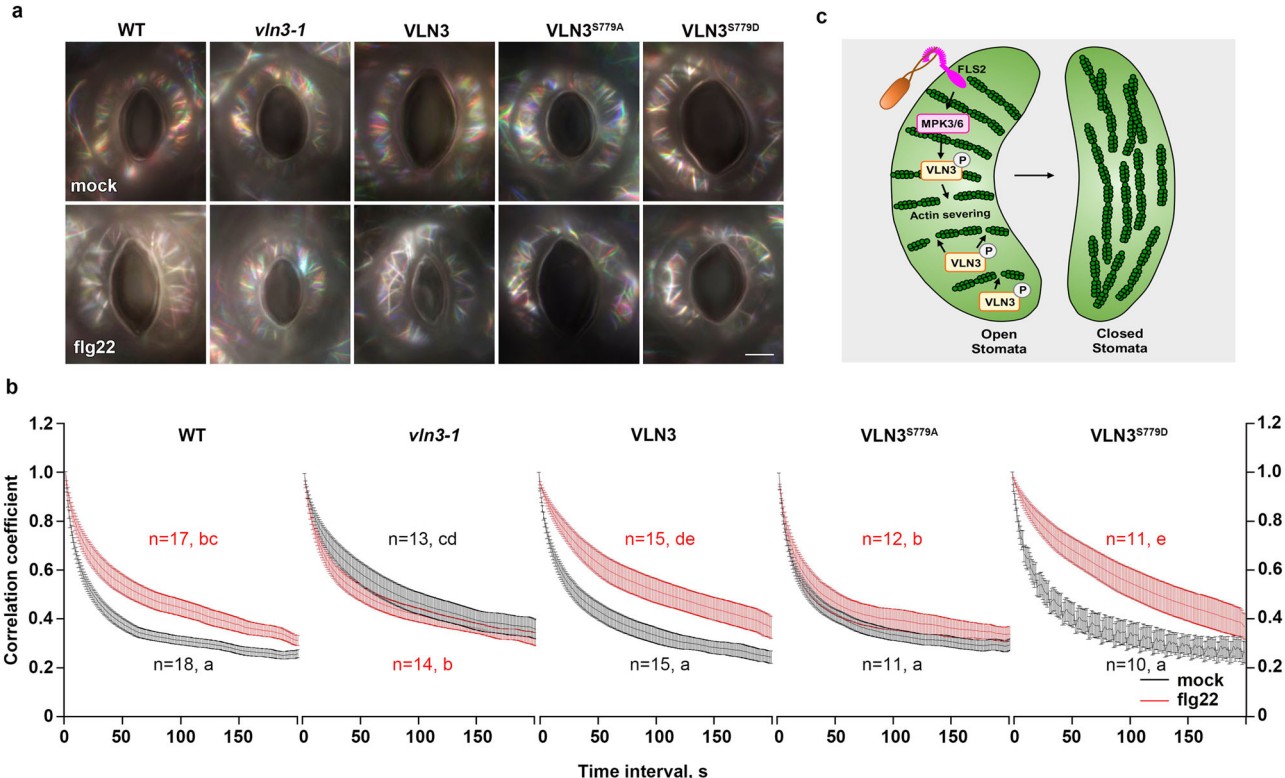

**Fig. 8 MPK3/6-mediated VLN3 phosphorylation is required for MAMP-induced actin dynamicity in guard cells. a** Merged images from time-lapse series were shown in indicated genotypes following mock or 10 μM flg22 treatments for 10 min. Scale bar, 5 μm. **b** Correlation coefficient analyses were performed on time-lapse series from WT, *vln3* mutant and complementary lines treated with mock or 10 μM flg22 for 10 min. Error bars represents SEM. Analyses were performed on time-lapse series taken from 10 leaves for each treatment and genotype. The sample sizes are indicated. Different letters indicate significant differences at $P < 0.05$, as determined by two-way ANOVA with Tukey's multiple comparisons test. Black or red letters indicate data from mock or flg22 treatment, respectively. The exact $p$ values are provided in the Source Data file. (**c**) A working model for the role of MPK3/6-VLN3 axis in regulating actin dynamics during stomatal immunity. Upon MAMP perception, VLN3 is phosphorylated by MPK3/6. This phosphorylation destabilizes the actin filaments, leading to radial actin array disruption in the open stomata, which facilitates actin reorganization to initiate MAMP-induced stomatal closure.

MPK3/6-VLN3 axis is required for actin reorganization during flg22-induced stomatal closure. The functional significance of VLN3 Ser779 phosphorylation was demonstrated by the findings that VLN3$^{S779A}$ fails to restore the defects in actin dynamics and stomatal defense in the *vln3* mutant, whereas wild-type and VLN3$^{S779D}$ do. Collectively, our data demonstrate that VLN3 phosphorylation by MPK3/6 regulates actin dynamics to activate stomatal defense (Fig. 8c).

Consistent with previous phosphoproteomic analyses, we confirmed that VLN3 is rapidly phosphorylated following flg22 activation[20,21]. Moreover, MPK3/MPK6 are identified as protein kinases responsible for this phosphorylation, and Ser779 is a critical phosphorylation site. In addition to MPK3/6, MPK4 is highly expressed in guard cells[33], which might also play a role in VLN3 phosphorylation during the stomatal defense. It has been suggested that VLN3 is a substrate of different protein kinases under various stress conditions[21]. Wang et al. (2020) showed that VLN3 is phosphorylated by CKL2, CRLK2, SOS2, SnRK2s, and CPK11 in vitro. All of the identified phosphosites of VLN3 are localized in a small region of linker domain, and different kinases seem to phosphorylate different sites[20,21]. These findings indicate a potential phosphorylation code for the regulation of VLN3 to mediate different responses triggered by different stresses. Here, we investigated how MPK3/6-induced phosphorylation affects the biochemical activity of VLN3. We showed that the abilities of VLN3 to bind to and bundle actin filaments are not altered by phosphorylation, whereas the Ca$^{2+}$-dependent severing activity is significantly enhanced. However, it remains unclear how phosphorylation at the linker region would have an impact on severing given that this domain predominately mediates actin bundle formation, and the core domain is responsible for the actin-severing properties[37]. Studies have demonstrated that there is an auto-inhibition mechanism in vertebrate villins to prevent actin severing. Tyrosine phosphorylation at the core domain releases the auto-inhibited conformation, promoting the actin-severing activity of villins[38,39]. This auto-inhibition mechanism may also exist in plant villins. We hypothesize that an association might exist between the linker and core domain, which leads to the auto-inhibited conformation of VLN3. Either calcium binding at core domain or phosphorylation at linker region could release this interaction, resulting in a conformational change in villin from an auto-inhibited state to an active state, thus exposing the actin-severing site. However, this hypothesis requires further investigation.

VLN3 and its closest homolog, VLN2, are expressed in all organs and various cell types[28,29]. Single mutants of these proteins do not result in any obvious developmental defects. Actin organization in epidermal cells of single mutants is similar to WT[29]. However, loss of both VLN2 and VLN3 results in fewer thick bundles and more thin bundles in the cortical actin array, leading to an impaired directional organ growth and sclerenchyma development[28,29]. In this study, we uncovered the function of VLN3 in guard cells. Under the unstimulated condition, actin filaments in *vln3* mutant cells are less frequently merged into bundles, but are turned over at a similar rate to that noted in WT cells. Consequently, the altered dynamic properties of actin filaments result in a more dense and less bundled cortical array. These data suggest that VLN3 plays a major role in actin bundle formation in guard cells. When stimulated with flg22, the *vln3* mutant fails to close stomata, whereas the single mutant of VLN2 or VLN4 shows a WT response. There is no additive effect in the *vln2vln3* double mutant, suggesting that VLN3 is the dominant villin during stomatal defense. It is well-recognized that actin array reorganization in guard cells is important for proper stomatal closure[6–8,10,13,34,40,41]. However, the mechanism underlying this structural transition remains to be determined. It

has been proposed that the destruction of existing actin networks in open stomata facilitate their closure[34,35]. Here, we showed that flg22 treatment stimulates changes in actin dynamics within minutes in guard cells, and the overall dynamicity of the actin array decreases significantly compared to mock treatment. Analyses of individual actin filaments revealed that flg22 treatments destabilize the actin array not only by preventing bundles formation but also by triggering faster filament turnover. These combined effects contribute to the disassembly of the radial actin array in the open stomata and promote stomatal closure. Additionally, our genetic results showed that the actin dynamics fail to respond to MAMP in *vln3* mutant guard cells, demonstrating the requirement of VLN3 for actin disassembly to initiate flg22-induced stomatal closure.

In plant cells, MAMP perception by cell surface receptors results in rapid cellular signaling events that occur on timescales of seconds to minutes, including cytosolic Ca$^{2+}$ fluxes, activation of MAPK and CDPK cascades, and accumulation of ROS and signaling phospholipids. These fast signals could alter the actin cytoskeleton by regulating the activity of actin-binding proteins[5]. In this study, we showed that MPK3/6 function upstream of the actin cytoskeleton. In guard cells, loss of MPK3/6 reduces the extent of actin filament bundling and increases the filament density. Moreover, the actin array in the mutants is predominantly longitudinal. During the stomatal defense, MPK3/6 is required for the rapid disassembly of the radial actin array to close stomata. Actin cytoskeleton regulation by MPK3/6 is mediated by VLN3 phosphorylation. In vitro, MPK3/6-mediated VLN3 phosphorylation enhances actin turnover. In vivo, inhibiting this phosphorylation fails to trigger MAMP-induced actin rearrangements. Moreover, constitutive VLN3 phosphorylation activates stomatal defense in the absence of upstream kinases. Thus, we provided genetic evidence that MPK3/6 phosphorylate VLN3 to initiate stomatal immunity by regulating actin dynamics.

Actin networks in different cell types respond distinctly to innate immune activation. In leaf pavement cells or hypocotyl epidermal cells, MAMP treatment increases actin filament abundance and enhances the overall actin array dynamicity[16,17,42,43]. Detailed investigations on dynamic behaviors of actin filaments suggest that MAMP-induced increases in actin abundance result from reduced filament disassembly, as well as increased availability of barbed ends for actin polymerization. In guard cells, however, actin density does not increase and the dynamicity of the actin array decreases after MAMP treatment[11,44]. Moreover, MAMP perception enhances the rate of actin filament turnover. The differences in actin responses between the epidermal cells and guard cells demonstrate that the architecture and properties of actin arrays that support different defense responses are highly unique. This notion is further supported by the findings that, unlike CPK3[44], the MPK3/6-VLN3 axis is not involved in actin responses in pavement cells following flg22 perception. These data suggest that innate immune signals that act on the actin cytoskeleton and response regulators vary between different cell types. The repertoire of mechanisms that control actin filament dynamics in vivo is more complex than previously appreciated.

## Methods

**Plant materials and growth conditions**. Arabidopsis plants used in this study include Col-0, *vln2-1* (SAIL_813_H02), *vln3-1* (SALK_117097), *vln3-2* (SALK_078340), *vln2vln3*[28,29], *vln4-1* (SALK_049058), MPK3SR line #64 and MPK6SR line #58[31]. The plants were grown in the growth room at 23 °C at 70% relative humidity with a 10-/14-h day/night photoperiod.

To generate VLN3 complementary lines, the native *VLN3* promoter that was 2,299 bp in length[29] and cDNA were amplified and cloned into pCAMBIA1300. Desired VLN3 mutant plasmids were generated by site-directed mutagenesis, and

the resulting constructs were introduced into *vln3-1* plants by *Agrobacterium*-mediated transformation.

**Disease assay**. To measure bacterial growth on *Arabidopsis*, 24-day-old leaves were hand infiltrated with bacterial suspensions at $1 \times 10^5$ colony-forming units per ml using a needless syringe or dip inoculated $5 \times 10^5$ colony-forming units per ml. Two leaf discs with a diameter of 0.5 cm$^2$ were collected at 48 h after infection and ground in 10 mM $MgCl_2$. Following bacterial recovery, serial dilution of leaf extracts was performed. A 2-μl drop from each dilution was plated for counting bacterial colonies. Each data points represents average bacterial numbers from three replicates[16].

For pathogen entry assays, detached leaves were illuminated for 2.5 h before 1-h treatment with DC3000 (OD = 0.5). Leaves were washed by 0.02% Silwet L-77 for 10 s. Pathogen entry was measured by directly counting colonies.

**Stomatal aperture assay**. Leaf peels were collected from the abaxial side of 3- to 4-week-old plant leaves, and incubated in stomata-opening buffer (50 mM KCl, 10 mM $CaCl_2$, and 10 mM MES, pH 6.15) in a growth chamber at 23 °C under constant illumination. Stomatal apertures were measured after treatment with mock, 10 μM flg22 or DC3000 (OD = 0.1) for 1 h using ImageJ software.

**PTI response analyses**. For MAPK activation assay, 10-d-old seedlings were soaked in water containing flg22 and frozen in liquid nitrogen. Total protein was extracted using the extraction buffer (50 mM HEPES, pH 7.5, 150 mM KCl, 1 mM EDTA, 0.5% Triton X-100, 1 mM DTT, proteinase inhibitor cocktail). Protein extracts were probed with anti-MPK6 (Sigma-Aldrich) or anti-p44/42 MAPK (Cell Signaling Technology) to assess MPK6 protein levels or phosphorylation of MPK3, MPK4, and MPK6. For the oxidative burst assay, leaf disks (4-mm diameter) from 4-week-old plants were floated on water overnight in a 96-well plate (one disk per well) and then treated with 1 μM flg22 in 100 μl buffer containing 20 μM luminol and 10 μg ml$^{-1}$ horseradish peroxidase. The luminescence was recorded using a FlexStation3 (Molecular Devices). For callose deposition analyses, 2-week-old seedlings were treated with 1 μM flg22 for 24 h. To stain callose, seedlings were incubated for at least 24 h in 95 to 100% ethanol until all tissues were transparent, were washed in 0.07 M phosphate buffer (pH = 9), and were incubated for 1 to 2 h in 0.07 M phosphate buffer containing 0.01% aniline-blue (Sigma-Aldrich)[45]. Images of callose deposits from whole cotyledons were collected using a Zeiss ImagerM1 epifluorescence microscope equipped with a 20×0.5-numerical aperture PlanFluor objective under ultraviolet light. The number of callose spots was quantified using Image J.

**Split-luciferase complementation assay**. To generate constructs of VLN3-Cluc, MPK3-Nluc, and MPK6-Nluc, the cDNAs were amplified and cloned into Cluc-pCAMBIA1300 or Nluc-pCAMBIA1300. *Agrobacterium tumefaciens* GV3101 containing the indicated plasmids was infiltrated into expanded leaves of *N. benthamiana* and incubated in the growth room for 48 h before the LUC activity measurement. For the CCD imaging and LUC activity measurement, 1 mM luciferin was sprayed onto the leaves. The cooled CCD imaging apparatus was used to capture the LUC image[46].

**Coimmunoprecipitation assay**. Full-length cDNAs of *MPK3* and *MPK6* were cloned into the Flag tag and *VLN3* was cloned into the GFP tag to generate *35 S:Flag-MPK3, 35 S:Flag-MPK6,* and *35 S:GFP-VLN3*. These resulting constructs were infiltrated into *N. benthamiana* via the *Agrobacterium*-mediated method. After 48 h, total proteins were extracted for coIP with the extraction buffer and incubated with anti-Flag agarose beads for 2 h. Beads were washed five times with washing buffer (50 mM HEPES, pH 7.5, 150 mM KCl, 1 mM EDTA, 0.5% Triton X-100, 1 mM DTT). After washing, the immunoprecipitated proteins were separated by SDS-PAGE and detected by anti-GFP and anti-Flag immunoblot.

**Protein purification**. The cDNAs of wild-type, mutated VLN3, its N terminus (719 amino acids) and C terminus (247 amino acids) were cloned into the pGEX-6p-1 vector. The resulting vectors were transformed into *E. coli* (strain BL21). The recombinant proteins were purified with glutathione sepharose.

**Phosphorylation assay and phosphosite identification**. In the in vitro kinase assay, recombinant His-MPK3 and His-MPK6 (0.3 μg) were activated by incubation with MKK5$^{DD}$ (0.1 μg) in reaction buffer (25 mM Tris-HCl pH 7.5, 10 mM $MgCl_2$, 50 μM ATP, and 1 mM DTT) in a 10-μl total volume at 30 °C for 0.5 h. Activated MPK3 and MPK6, or MPK4$^{CA}$ were then incubated with GST-VLN3 or its mutants (1.5 μg) in the reaction buffer (25 mM Tris-HCl pH 7.5, 10 mM $MgCl_2$, 50 μM ATP, 1 mM DTT, and 1 μCi [γ-$^{32}$P] ATP) in a 25-μl total volume at 30 °C for 0.5 h. The reaction was terminated with 6 × SDS loading buffer. After incubation at 100 °C for 5 min, the reaction products were separated on 12% SDS-PAGE and stained using Coomassie Brilliant Blue R 250[47]. The phosphorylation signals were detected by a Typhoon 9410 phosphor imager. Phosphorylation signals were quantified using ImageJ software. To determine which region of VLN3 is phosphorylated by MPK3/6, N-VLN3 and C-VLN3 was used in the kinase assay.

After phosphorylation, proteins were separated in phos-tag gel (10% SDS-PAGE, 50 μM Phos-tag, 100 μM $MnCl_2$). After electrophoresis, the gel was incubated in the transfer buffer (50 mM Tris, 40 mM glycine) containing 10 mM EDTA thrice, washed in transfer buffer for 10 min, and then transferred to a nitrocellulose membrane. Then, N-VLN3 and C-VLN3 was detected with the anti-GST antibody.

To detect VLN3 phosphorylation in plants, HA-tagged N-VLN3 and C-VLN3 were expressed in Arabidopsis protoplasts. Following treatment with mock or 100 nM flg22 for 10 min, total proteins were extracted with extraction buffer, separated in a phos-tag gel and detected by immunoblot analysis.

For phosphosite identification, phosphorylated C-VLN3 by MPK3/MPK6 in vitro was excised from a SDS-PAGE gel, and in-gel digestion was performed using a well-established protocol with slight modifications[48]. Briefly, the protein embedded in-gel slices was reduced with 10 mM DTT, alkylated with 55 mM iodoacetamide, and then digested overnight with sequencing grade trypsin (Sigma-Aldrich) at 37 °C. The tryptic peptides were analyzed by LC-MS/MS using nanoLC-LTQ-Orbitrap XL (ThermoFinnigan). Peptide identification and phosphosites assignments were performed with the Proteome Discoverer software (version 1.4, Thermo Fisher). The *Arabidopsis thaliana* proteome sequences (Uniprot) were used as the database and the mass tolerances were set to 20 PPM for precursor and 0.6 Da for fragment ions for the database search.

Phosphosite-specific antibodies were custom-made by Abmart. Briefly, a phosphorylated Ser779 peptide [SGRTS(pS)PSRD] and a control peptide [SGRTSSPSRD] were synthesized, and rabbits were immunized with the phosphopeptide conjugated to keyhole limpet hemocyanin (KLH) carrier. The polyclonal antiserum was purified by affinity chromatography using a phosphopeptide, and the eluate was passed through the column coupled with control peptides to remove nonspecific antibodies.

**Biochemical characterization of phosphorylated VLN3**. High-speed cosedimentation assays were used to determine the actin-binding and depolymerizing activities of VLN3. Low-speed cosedimentation assays were used to determine the actin-bundling activity[37]. Actin was dialyzed overnight against buffer G (5 mM Tris-HCl, 0.1 mM $CaCl_2$, 1 mM $NaN_3$, 0.2 mM ATP·2Na, 0.5 mM DTT, pH 7.0). Before use, VLN3 and actin were clarified by centrifugation at 55,000 g for 1 h. Actin was prepolymerized in 1× KMEI buffer (50 mM KCl, 1 mM $MgCl_2$, 1 mM EGTA, 10 mM imidazole, pH 7.0) at room temperature for 2 h. VLN3 was phosphorylated using an in vitro kinase assay. Recombinant MPK3 and MPK6 were activated by incubation with MKK5$^{DD}$ in the reaction buffer (20 mM Tris-HCl, pH 7.5, 10 mM $MgCl_2$, 50 mM ATP and 1 mM DTT) at 30 °C for 0.5 h. Activated MPK3 and MPK6 were used to phosphorylate recombinant GST-VLN3 (1:10 enzyme-substrate ratio) in the reaction buffer at 30 °C for 30 min. Phosphorylated VLN3 was then incubated with preformed actin filaments at 1:3 ratio in a 100-μl reaction volume. Low-speed cosedimentation assays were performed in the presence of 2 mM EGTA, and 200 μM free $Ca^{2+}$ was added in high-speed cosedimentation assays. Following incubation, the reaction mix was centrifuged for 1 h at 55,000 g (high speed) or at 13,500 g (low speed). The supernatant (80 μl) was transferred to a separate tube, and 16 μl of 6× protein loading buffer was added to it. The pellet was suspended by 100 μl buffer G, and 20 μl of 6 × protein loading buffer was added. The samples were then separated by 12% SDS-PAGE and stained with Coomassie Brilliant Blue R (Sigma-Aldrich). Individual severing events along actin filaments were imaged by time-lapse TIRF microscopy[49]. The assembly of monomeric actin (Oregon-green labeled) was initiated by the addition of one-tenth volume of 10 × KMEI (0.5 mM KCl, 10 mM $MgCl_2$, 10 mM EGTA, 0.1 mM Imidazole, pH 7.0). Actin filaments (15~50% Oregon-green-actin, 0.125~1 mM) were mixed with 2 × TIRF buffer (20 mM imidazole [pH 7.4], 100 mM KCl, 2 mM $MgCl_2$, 2 mM EGTA, 0.4 mM ATP, 10 mM DTT, 30 mM glucose, 40 mg ml$^{-1}$ catalase, 200 μg ml$^{-1}$ glucose oxidase, and 1% methylcellulose) and transferred to a microscope flow chamber for imaging at room temperature. VLN3 protein was introduced after placing the chamber on the microscope stage. TIRF images were collected at 1- to 2-s intervals using ELYRA 7 (Carl Zeiss). Microscope slides (24 × 50 mm 12-545-F; Fisher Scientific) and cover-slips (22 × 22 mm 12-542-B; Fisher Scientific) were cleaned using piranha solution (a 3:1 mixture of sulfuric acid and 30% hydrogen peroxide) overnight. The glass then was incubated with 2 mg ml$^{-1}$ methoxy-PEG-silane MW 5000 and 2 mg ml$^{-1}$ biotin-PEG-silane MW 3400 (Laysan Bio) in 95% ethanol (pH 2.0) at 70 °C. Two parallel strips of double-sided tape were placed on both ends of the coverslip to create a flow chamber.

**Quantitative analysis of actin dynamics in guard cells**. To visualize actin filaments in plant cells, the *35 S:GFP-lifeAct* construct was transformed into the wild type, *vln3* mutants, VLN3 complementary lines and *MPK3SR* lines. Guard cells from 10-d-old plants were used for live-cell imaging. Actin organization analyses were performed on images acquired with a PerkinElmer UltraView Vox spinning disk microscope equipped with 60×1.42 Numerical Aperture objective. For time-lapse imaging of actin filament dynamics, the actin array in guard cells was recorded by variable angle epifluorescence microscopy (VEAM) at 1-s intervals, and a series of 200 images were collected. To analyze the bundling and severing frequency in guard cells, we selected a region of interest (ROI; 36-μm$^2$) to perform the analysis. The dynamics of actin filaments in this region was tracked in a consecutive 100 s and analyzed. The bundling and severing frequencies were defined as the number of these events per unit area per unit time (events μm$^{-2}$ s$^{-1}$)[37]. For correlation

coefficient analyses, time-lapse VAEM series were cropped and analyzed using the built-in MATLAB function corr2 defined by Vidali and colleagues[16,36].

**Reporting Summary**. Further information on research design is available in the Nature Research Reporting Summary linked to this article.

## Data availability

The authors declare that all data supporting the findings of this study are included in the manuscript and its supplementary files are available from the corresponding author upon request. Source data are provided with this paper.

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

## Acknowledgements

We thank Dr. Juan Xu (Zhejiang University) for generously providing *MPK3SR* and *MPK6SR* lines. MPK4$^{CA}$ was kindly provided by Dr. Juan Xu (Zhejiang University) and Dr. Jin-Long Qiu (Chinese Academy of Sciences). We thank Dr. Lei Zhu (China Agriculture University) for helping with the in vitro kinase assay. We thank Dr. Tonglin Mao (China Agriculture University) for helpful discussion on the manuscript and Xiang Ding (Laboratory of Proteomics, Institution of Biophysics, Chinese Academy of Sciences) for mass spectrometry data analysis. We are grateful to Dr. Chao Xi and Dr. Jin Liu from the Experimental Technology Center for Life Sciences, Beijing Normal University for technical support. This research was funded by the National Natural Science Foundation of China (31801135, 92054101).

## Author contributions

M.Z., M.G., and J.L. designed the experiments. M.Z., M.G., Z.Z., B.W., Q.P., and J.L. performed experiments. M.Z., M.G., J.-M. Z., and J.L. analyzed the data; M.Z., J.-M. Z., and J.L. wrote the paper. All authors commented and agreed on the manuscript before submission.

## Competing interests

The authors declare no competing interests.
