## [Peer Review File · Nature Communications]

MPK3- and MPK6-mediated VLN3 phosphorylation regulates actin dynamics during stomatal immunity in ArabidopsisREVIEWER COMMENTS

Reviewer #1 (Remarks to the Author):

Review of Phosphorylation of an Arabidopsis villin activates host actin remodeling to 2 restrict microbial invasion by Zou et al.

When I accepted the review, I was expecting a manuscript with a very different scope that would focus on actin polarization in regular root or leaf epidermal cells. This is caused by the failure to mention stomatal immunity in the title, abstract and introduction. Instead, these parts focus on access to the host cytoplasm and cellular defense. In the presented research, the focus is on gaining access to internal tissues through stomata and not the invasion of cells. I would expect that an introduction for the presented work would focus on the mechanical barriers constructed by the plant epidermis and the stomata as weak spots in this line of defense by the necessity of gas exchange. The actin cytoskeleton in these cells plays a role in controlling stomatal aperture and the perceived presence of a pathogen triggers stomatal closure. This has very little relationship with the role of actin in limiting cellular invasion. The novelty of this work is the identification of a pathway from pathogen perception to actin mediated stomatal closure, and this should be the focal point of title, abstract and introduction. I suggest a rewrite of the introduction with a focus on stomatal immunity and relocation of the information about this process in the first part of the results to the introduction.

The findings presented in the manuscript are generally clear and sound, but I have some concerns about specific experiments.

Line 149-151: why were only C- and N-terminal fragments, and not full length VLN3 tested? Using full length VLN3 is a good control to test for correct folding of the fragments.

I have some concerns about the validity of the phosphorylation assays (fig 2). Firstly, overexpression constructs and in vitro assays are used to confirm interaction. Although the reduction of VLN3 phosphorylation in the mpk3/6 double helps, a response that occurs in stomata is studied in complete leaves. The pathway appears to be operational in a broad range of cells, but no evidence is given for this. Are MPK3/6 and VLN3 expressed in all cell types? Do the treatments reach all cell types? What is the function of the phosphorylation response in cells other than guard cells? In addition, differences in phosphorylation appear to be minor and should be validated by a proper quantification.

In the paragraph from line 251 it is shown that actin reorganization in guard cells upon flg22 treatment is disrupted in both the vln3 and mpk3/6 mutants in a similar way. An informative control would be to

test the *vln3/mpk3/6* mutant for the absence of additive defects. At the very end of this paragraph it is mentioned that a similar effect in actin reorganization is observed in pavement cells. A quantification of these results is given in figure S6, but I do not think this quantification is very informative by itself. It would be useful to include images showing how the actin organization is modified upon *flg22* treatment.

In figure 6, a clear decrease in actin bundling is shown when the MPK3/6 and VLN3 pathway is activated by *flg22* treatment. If the title of the paragraph that describes the results of figure 6, 'MPK3/MPK6-VLN3 axis regulates actin dynamics in guard cells upon *flg22* perception' is correct, the base level of actin bundling should be the same in the mutants and wt. This is not the case, the mutants display the low bundling levels even before treatment with *flg22*. The authors attempt to explain this by stating that issues in the orientation of the native array cause defects in array reorientation, but I do not see any evidence for this. I might be ignorant here, but if the actin array is involved in determining the aperture of stomata, and the actin array is disrupted in untreated cells, why are no defects in stomatal aperture observed in untreated mutants?

Minor issues:

I know that stomatal movement is used regularly in literature, but I do not like this term very much as it suggests motility. I would suggest changes in stomatal aperture, or stomatal opening and closing as alternatives.

Line 133-134: please formulate 'multiple factors' more clearly.

Line 138-146: it is not clear from the text if the described modifications occur throughout the plant or only in guard cells.

In supplemental figure 1, the location of the callose deposits is not mentioned either. Do callose deposits localize to stomata?

Figure S1d has extremely low contrast, please do not use pseudocolor but black and white to fix this.

The paper would benefit from a graphical summary in the last figure that incorporates the findings and regulation of stomatal aperture.

Using the term innate immunity gates for stomata does not cover their full function in gas exchange and could be considered as confusing.

Reviewer #2 (Remarks to the Author):

This article reports the phosphorylation of the villin protein, VLN3, by the mitogen-activated protein kinases MPK3/6. This phospho-regulation led to a reprogramming of the actin cytoskeletal structure, closure of stomates and forms part of the pre-invasive immune response to restrict bacterial entry into the intercellular spaces of plants. The finding is potentially interesting since it fills knowledge gaps of how bacteria (Melotto et al 2006) or specifically how MPK3/6 regulate stomatal immunity (Su et al 2017).

Nevertheless, I have several questions concerning the phosphorylation, clarity of the actin assays and the inoculation methods.

Phosphorylation:

1. Line 187: The authors mutated T777 and S779 simultaneously (and not “OR”, as stated in this line). Importantly, T777 is not a canonical MAPK-type of phosphosite. This questions its relevance, and all the subsequent functional analysis should have been performed with S779 alone and not both sites. (This is a major point to address since their claim in this article is that MPK3/6 regulate stomatal immunity via phosphorylation at these sites, which would be an inaccurate statement if one of them is not phosphorylated by MPK3/6).
2. In this context, mass spectrometric data for the phosphosite mapping (Fig S2A) should include more details (in particular, the confidence scores for phosphosite site annotation, the MS/MS spectra), for readers to assess the accuracy of site annotation.
3. Line 216: Is there any validation that the “phosphorylated VLN3” are indeed phosphorylated for these severing assays (Is the reaction buffer suitable for kinase reactions? e.g. contains ATP?). Furthermore, in the various kinase assays, there is no control with just the MKK5DD kinase alone. Can this also phosphorylate VLN3 directly (e.g. possibly at T777)?

4. Line 149/Fig 2: To verify VLN3 phosphorylation in vivo, they expressed truncated VLN3 fragments. While this enabled detection of phospho-modification of the C-terminal fragment, it would be physiologically more relevant to analyze the full length protein first.

5. Line 159 onwards: It is not entirely clear why the authors focussed exclusively on MPK3/MPK6. As mentioned by them, reference #22 (Berriri et al) showed that elevated MPK4 activity compromised response to sprayed bacteria (but not hand-infiltrated bacteria) and is hence involved in stomatal response. That paper also showed a strong, almost exclusive, expression of MPK4 (using pMPK4-MPK4:GFP transgenic lines) in the guard cells. There are also several other papers that report involvement of MPK4 in stomatal closure. It would be better to include MPK4 in this work, in part also to show specificity for their assays (e.g. those used in Fig 2B-E).

6. In this context, is the phosphoshift of VLN3 in the Phos-Tag analysis lost in the mpk3mpk6 mutant? (This would lend further support the anti-pS/pT-based assay, Fig 2G).

Actin severing assay:

7. Figure S3: Generally, a better description of the analysis performed will help readers (not familiar with this actin filament severing assay) to interpret the data. Furthermore, in Fig S3A, the second and third lanes have different outcome although they are labelled with identical conditions (mislabelled?).

8. In this context, shouldn't the phospho-mimetic of VLN3 have more severing activity (Fig S5C)? There seems to be no differences for the actin signals in the pellet fractions (between no VLN3, lane 1, and adding the various VLN3 variants, lanes 2-4).

Inoculation method:

9. For validating their hypothesis on stomatal defense (in Figure 8), they only tested spray inoculation to compare the VLN phospho-variants. However, Fig 1A and B also demonstrated that the vln3 knock-out mutants are also compromised in immunity after hand-infiltration. Here, it will be prudent to additionally assess susceptibility of the complemented lines by bacteria infiltration in parallel. Possibly, a "bacterial entry" assay (as in Fig 1F) should have been tested. Such a comparison between spray and infiltrated inoculation may provide a clearer idea of the contribution of stomatal immunity versus post-invasive immunity (when the bacteria have entered the intercellular spaces of the leaves).

Other points:

10. Please check the statistics tests used throughout the manuscript. For instance, Fig S1C should have been a 2-way ANOVA analysis to separate effects of genotype from treatment. In several figures, letters are used to denote statistically distinct groups but described as student's t-test (which is only pairwise test and not for multiple comparisons). Vice versa, another example is Fig 7B, which is stated as ANOVA in the figure legend but based on the description, is in fact only pair-wise comparisons (If this was indeed a Dunnett's ANOVA assay, please state it specifically as such).

11. Labelling in Fig 3B? (difference between black and orange columns?). Same issue for Fig S4B.

12. Fig. S2B: Composite/compiled figures should indicate spliced blots (e.g. with vertical line).

13. MAMP treatment labels missing in Fig S6.

14. Fig. 5. It would be useful to readers if they briefly described what parameters are being compared in the "correlation coefficient" analysis.

Reviewer #3 (Remarks to the Author):

The manuscript by Zou et al demonstrates an axis involves MPK3/MPK6-VLN3 in flg22-induced stomatal closure and stomatal immunity against Pst DC3000. In general, I feel the work is well designed and experiments are carefully performed. I have only one major concern as follows:

As is shown in Fig 2F and Supplemental Fig 2B, HA-C-VLN3T777A, S779A is still responsive to flg22 treatment. Though the highest band is abolished upon flg22 treatment in HA-C-VLN3T777A, S779A, the lowest band was reduced and the signal for the middle band was enhanced upon flg22 treatment. Thus, other phosphorylation sites also contribute to flg22-induced band shift of C-VLN3. This notion is supported by the data in Supplemental Fig 2B, where it shows that the signal of the middle band was enhanced in HA-C-VLN3T849A, S850A upon flg22 treatment.

As is shown in Fig 2E, MPK3 and MPK6 can still induce band shift of HA-C-VLN3T777A, S779A, i.e. there are other phosphorylation sites of VLN3 by MPK3/MPK6, possibly S850. A combination of further mutations should be tested.

Furthermore, if T777 and S779 are the only two phosphorylation sites by MPK3/MPK6 in response to flg22, a VLN3T777D, S779D version should constitutively regulate actin dynamics. However, this prediction was not observed in Fig 7A and B, and Fig 8A and B.

Taken together, I strongly suggest the authors further confirm the MPK3/MPK6 phosphorylation sites in VLN3 biochemically and physiologically.

MAPKs are proline-dependent kinases, a proline residue always follows the phosphorylation sites. It should be careful to conclude that T777 is a phosphorylation site of MPK3/MPK6. And this should be confirmed by comparing single T777A, S779A mutations and double T777A S779A mutation.

Minors:

Such as Line 113, “dominate” should be “dominant”.

REVIEWER COMMENTS

In response to Reviewer #1's comments:

Review of Phosphorylation of an Arabidopsis villin activates host actin remodeling to 2 restrict microbial invasion by Zou et al.

When I accepted the review, I was expecting a manuscript with a very different scope that would focus on actin polarization in regular root or leaf epidermal cells. This is caused by the failure to mention stomatal immunity in the title, abstract and introduction. Instead, these parts focus on access to the host cytoplasm and cellular defense. In the presented research, the focus is on gaining access to internal tissues through stomata and not the invasion of cells. I would expect that an introduction for the presented work would focus on the mechanical barriers constructed by the plant epidermis and the stomata as weak spots in this line of defense by the necessity of gas exchange. The actin cytoskeleton in these cells plays a role in controlling stomatal aperture and the perceived presence of a pathogen triggers stomatal closure. This has very little relationship with the role of actin in limiting cellular invasion. The novelty of this work is the identification of a pathway from pathogen perception to actin mediated stomatal closure, and this should be the focal point of title, abstract and introduction. I suggest a rewrite of the introduction with a focus on stomatal immunity and relocation of the information about this process in the first part of the results to the introduction.

AU response: We rewrote the title, abstract and introduction to be more focused on stomatal immunity, as suggested by the reviewer.

The findings presented in the manuscript are generally clear and sound, but I have some concerns about specific experiments.

Line 149-151: why were only C- and N-terminal fragments, and not full length VLN3 tested? Using full length VLN3 is a good control to test for correct folding of the fragments.

AU response: The mobility-shift of full-length VLN3 was subtle after flg22 treatment (Supplemental Fig. 2a). This may due to large phosphoproteins are often poorly

resolved in phos-tag assays (Bi et al., 2018). We thus used truncated VLN3 fragments to detect in-vivo phosphorylation.

In updated Fig. 2d,e, the in vitro kinase assays were now performed using full-length VLN3. We also generated antibodies that specifically recognizes pSer779, and by immunoblotting with this antibody, we confirmed that full-length VLN3 is phosphorylated upon flg22 activation and MPK3/6 is required for this phosphorylation (see updated Fig. 2h).

I have some concerns about the validity of the phosphorylation assays (fig 2). Firstly, overexpression constructs and in vitro assays are used to confirm interaction. Although the reduction of VLN3 phosphorylation in the mpk3/6 double helps, a response that occurs in stomata is studied in complete leaves. The pathway appears to be operational in a broad range of cells, but no evidence is given for this. Are MPK3/6 and VLN3 expressed in all cell types? Do the treatments reach all cell types? What is the function of the phosphorylation response in cells other than guard cells? In addition, differences in phosphorylation appear to be minor and should be validated by a proper quantification.

AU response: Based on the published gene expression data, it was shown that MPK3/6 and VLN3 are expressed in all organs (Bush and Krysan, 2007; Winter et al., 2007; Bao et al., 2012; van der Honing et al., 2012). Bush et al. (2007) showed that MPK6 are expressed in whole leaves (Bush and Krysan, 2007). Here, we also observed VLN3 expression in leaves using *pVLN3:VLN3-GFP/vln3* transgenic lines. As shown in Supplemental Fig. 1d, VLN3-GFP signals was observed in both leaf epidermal pavement cells and guard cells. However, we can't determine whether VLN3 is also expressed in mesophyll cells because it is difficult to obtain GFP signals from inside the leaf tissue.

The flg22 treatments in updated Fig. 2 were performed by adding flg22 peptide into the suspension of protoplasts that were prepared from whole rosette leaves, we think the treatment should reach all leaf cell types by this method.

In Fig. 3b and Supplemental Fig. 4a,b, we investigated bacterial growth on WT and transgenic lines of *vln3* mutant expressing VLN3, phospho-null or phosphomimic VLN3 derivatives. The data showed that phospho-null VLN3 derivatives failed to restored the bacterial susceptibility of *vln3* mutant to WT level in both spray-inoculation (Fig. 3b; Supplemental Fig. 4a) and hand-infiltration

(Supplemental Fig. 4b) assay. In addition, the level of flg22-induced callose deposition in phospho-null VLN3 transgenic lines is comparable to that in *vln3* mutant (Supplemental Fig. 4c). These data suggest that MPK3/6-mediated VLN3 phosphorylation is operational in other leaf cell types, the phosphorylation response is required for both stomatal and apoplastic defenses.

To validate the phosphorylation of VLN3 by MPK3/6, we conducted in vitro kinase assay with recombinant MPK3/6 and full-length VLN3, using MPK5^{DD} and MPK4 as controls. The phosphorylation signal was detected by [γ -³²P]ATP autoradiography (see Fig. 2d). The data suggest that VLN3 is specifically phosphorylated by MPK3/6. We also performed the kinase assay on different VLN3 mutant proteins with Ser substituted with Ala, and quantified the pSer band intensity of these proteins. As shown in Fig. 2e, phosphorylation of VLN3 is greatly reduced with the Ser779 single substitution, suggesting that Ser779 is required for MPK3/6-mediated VLN3 phosphorylation.

To confirm the requirement of MPK3/6 for VLN3 phosphorylation in vivo, we performed two experiments. First, we expressed VLN3's C-terminus in WT and *mpk3/6* double mutant protoplasts. As shown in Fig. 2g, the motility shift of VLN3-C was abolished in *mpk3/6* double mutants compared to WT (see Fig. 2g). We further developed an antibody recognizing phosphorylated VLN3 S779 (α -pS779). The pS779 antibody specifically detects VLN3 phosphorylation by MPK3/6, whereas the signal was absent in the controls with MPK4 or MKK5^{DD} (Supplemental Fig. 3b). Anti-pS779 cannot detect the VLN3 S779A variant (Supplementary Fig. 3c). Immunoblotting with anti-pS779 revealed that VLN3 S779 was unphosphorylated in nonstimulated WT but became strongly phosphorylated upon stimulation with flg22 (Fig. 2h), indicating that this residue is specifically phosphorylated in a flg22-dependent manner. In contrast, the Ser779 phosphorylation was abolished in *mpk3/6* double mutants (Fig. 2h), indicating that MPK3/6 are indeed required for the flg22-induced phosphorylation of VLN3 Ser779 in the plant cell. Taken together, these results support that MPK3/6 directly phosphorylates VLN3 at Ser779 both in vitro and in vivo.

In the paragraph from line 251 it is shown that actin reorganization in guard cells upon flg22 treatment is disrupted in both the *vln3* and *mpk3/6* mutants in a similar

way. An informative control would be to test the *vln3/mpk3/6* mutant for the absence of additive defects. At the very end of this paragraph it is mentioned that a similar effect in actin reorganization is observed in pavement cells. A quantification of these results is given in figure S6, but I do not think this quantification is very informative by itself. It would be useful to include images showing how the actin organization is modified upon flg22 treatment.

AU response: We generated *vln3/mpk3/6* triple mutant and performed quantitative analyses on *vln3* single, *mpk3/6* double and *vln3/mpk3/6* triple mutant. As shown in Fig. 5f, there is no additive defect in flg22-induced actin responses in the triple mutant compared with *mpk3/6* double mutant, supporting that MPK3/6 and VLN3 function with the same immune signaling pathway (see Fig. 5).

Representative images showing actin response following flg22 treatment in pavement cells are now included, as suggested by the reviewer (see Supplemental Fig. 8).

In figure 6, a clear decrease in actin bundling is shown when the MPK3/6 and VLN3 pathway is activated by flg22 treatment. If the title of the paragraph that describes the results of figure 6, 'MPK3/MPK6-VLN3 axis regulates actin dynamics in guard cells upon flg22 perception' is correct, the base level of actin bundling should be the same in the mutants and wt. This is not the case, the mutants display the low bundling levels even before treatment with flg22. The authors attempt to explain this by stating that issues in the orientation of the native array cause defects in array reorientation, but I do not see any evidence for this. I might be ignorant here, but if the actin array is involved in determining the aperture of stomata, and the actin array is disrupted in untreated cells, why are no defects in stomatal aperture observed in untreated mutants?

AU response: We agree with the reviewer that MPK3/6 and VLN3 are not only required for actin reorganization during flg22-induced stomatal closure, but also play critical roles in maintaining the normal actin arrays in unstimulated guard cells. We modified the title of this paragraph to include this (see highlighted sentences, line 336-337 and line 388-389).

Data from this study and others showed that defect in native actin array (orientation, stability, dynamics, organization, etc....) have no effect on the steady state stomatal aperture under optimal stomatal opening conditions in the absence of

stress stimulus (Zhao et al., 2011; Jiang et al., 2012; Guo et al., 2016; Zheng et al., 2019), suggesting that changes in guard cell actin dynamics are necessary but not sufficient to trigger stomatal closure. In the presence of stress signals, the reorganization of actin cytoskeleton is important for proper stomatal closure. It still remains unclear how this reorganization is achieved during stomatal closure, it has been suggested that the stable open-aperture cable configuration is behaving as a cytoskeletal brake to stress-mediated guard cell closure, the radial-actin cable structures in open stomata need to be removed or reorganized to ensure proper stomatal closure induced by external signals. Therefore, actin filament destabilization is a necessary aspect of this process (Jiang et al., 2012; Guo et al., 2016). Our results showed that MPK3/6-mediated VLN3 phosphorylation occur within minutes after flg22 treatment (Fig. 2a) and the phosphorylation leads to enhanced severing (Figure 4; Supplemental 7). Based on this, we reasoned that MPK3/6 and VLN3 might be involved in the disassembly of the radial-actin array. We showed that after 10-min flg22 treatment, the actin filaments in WT cells become less stable and are quickly turned over (indicated by reduced bundling frequency and increased severing frequency, respectively), indicating that these behaviors may lead to flg22-induced rearrangement or disassembly of obstructive actin array, which ultimately facilitate stomatal closure. In *mpk3/6* and *vln3* mutant, these parameters were less responsive to flg22 than those in WT, suggesting that MPK3/6 and VLN3 are involved in actin behaviors induced by flg22. Therefore, the actin array in the open mutant stomata fail to disassemble, and remain as a stable open-aperture cable configuration, which consequently result in impaired stomatal closure in response to flg22. We modified the texts to help reader with data interpretation (line 388~445).

Minor issues:

I know that stomatal movement is used regularly in literature, but I do not like this term very much as it suggests motility. I would suggest changes in stomatal aperture, or stomatal opening and closing as alternatives.

AU response: Modified accordingly as suggested by the reviewer.

Line 133-134: please formulate 'multiple factors' more clearly.

AU response: Fixed (highlighted line 145~146).

Line 138-146: it is not clear from the text if the described modifications occur throughout the plant or only in guard cells.

AU response: Callose deposition, MAPK activation and ROS production were studied on whole seedling or rosette leaves. Experimental details were now included in Method (see updated method section, highlighted line 603~618).

In supplemental figure 1, the location of the callose deposits is not mentioned either. Do callose deposits localize to stomata?

AU response: The callose deposits from the whole leaf area are quantified and we stated this in updated Method (see updated method section, highlighted line 615~618). The localization of callose deposit within leaf tissue has not been reported yet. Using standard protocol, images of callose deposits are usually collected with epifluorescence microscope with 20x objective. With this magnification power, it is difficult to specify their locations. Thus, we used 63x objectives to collect images from different optical sections using confocal microscope. As shown in the figure below, after flg22 treatment, callose deposits are mainly localized at 10 μm to 30 μm below leaf abaxial surface, we fail to detect callose deposits in guard cells, suggesting that calloses are produced mainly from inner leaf tissue, possibly by mesophyll cells.

Figure S1d has extremely low contrast, please do not use pseudocolor but black and white to fix this.

AU response: Fixed (see Supplemental Fig. 1e)

The paper would benefit from a graphical summary in the last figure that incorporates the findings and regulation of stomatal aperture.

AU response: We agree, the graphical summary figure was now added (see Fig. 8c)

Using the term innate immunity gates for stomata does not cover their full function in gas exchange and could be considered as confusing.

AU response: We changes this accordingly as suggested by the reviewer.

In response to Reviewer #2's comments:

This article reports the phosphorylation of the villin protein, VLN3, by the mitogen-activated protein kinases MPK3/6. This phospho-regulation led to a reprogramming of the actin cytoskeletal structure, closure of stomates and forms part of the pre-invasive immune response to restrict bacterial entry into the intercellular spaces of plants. The finding is potentially interesting since it fills knowledge gaps of how bacteria (Melotto et al 2006) or specifically how MPK3/6 regulate stomatal immunity (Su et al 2017).

Nevertheless, I have several questions concerning the phosphorylation, clarity of the actin assays and the inoculation methods.

AU response: We thank Reviewer #2 for their interest in our work, and also for the thoughtful critiques below.

Phosphorylation:

1. Line 187: The authors mutated T777 and S779 simultaneously (and not "OR", as stated in this line). Importantly, T777 is not a canonical MAPK-type of phosphosite. This questions its relevance, and all the subsequent functional analysis should have been performed with S779 alone and not both sites. (This is a major point to address since their claim in this article is that MPK3/6 regulate stomatal immunity via

phosphorylation at these sites, which would be an inaccurate statement if one of them is not phosphorylated by MPK3/6).

AU response: We thank the reviewer to point this out. All functional analysis were now performed with Ser779 alone (see updated MS).

2. In this context, mass spectrometric data for the phosphosite mapping (Fig S2A) should include more details (in particular, the confidence scores for phosphosite site annotation, the MS/MS spectra), for readers to assess the accuracy of site annotation.

AU response: Mass spectrometric data were now added as suggested by the reviewer (see Supplemental Fig. 2c).

3. Line 216: Is there any validation that the “phosphorylated VLN3” are indeed phosphorylated for these severing assays (Is the reaction buffer suitable for kinase reactions? e.g. contains ATP?). Furthermore, in the various kinase assays, there is no control with just the MKK5^{DD} kinase alone. Can this also phosphorylate VLN3 directly (e.g. possibly at T777)?

AU response: The in vitro kinase assay was performed to phosphorylate VLN3 prior to the actin-related assays. The resulting mix from kinase assay was then added into the reactions in the high-/low-speed cosedimentation assay and TIRF assay. We took two approaches to make sure that VLN3 used in the actin assays are indeed phosphorylated. First, we performed a parallel kinase assay using C-VLN3, and confirmed its phosphorylation by phos-tag gel analyses (data not shown; full-length VLN3 are poorly resolved in phos-tag assays [Supplemental Fig. 2a]). We repeated these parallel experiments for several times, and continued to use the same experimental setups for VLN3 phosphorylation. As a second, reaction mix in the high-speed cosedimentation assay was probed with α -pS779 to validate VLN3 phosphorylation (Supplemental Fig. 6d).

Reaction buffer in high-speed cosedimentation assay and TIRF assay all contain ATP. Descriptions of high-/low-speed cosedimentation assays were now included in the Method section (see updated Method, highlighted line 692~711). Controls with just the MKK5^{DD} kinase alone were now added in the kinase assays and actin-related assays (see updated Fig. 2d; Fig. 4; Supplemental Fig. 7). We

cannot detect VLN3 phosphorylation by MKK5^{DD} alone (Fig. 2d). Moreover, incubation with MKK5^{DD} had no effect on the biochemical activity of VLN3 (Fig. 4; Supplemental Fig. 7).

4. Line 149/Fig 2: To verify VLN3 phosphorylation in vivo, they expressed truncated VLN3 fragments. While this enabled detection of phospho-modification of the C-terminal fragment, it would be physiologically more relevant to analyze the full length protein first.

AU response: The mobility-shift of full-length VLN3 was subtle after flg22 treatment (Supplemental Fig. 2a). This may be due to large phosphoproteins are often poorly resolved in phos-tag assays (Bi et al., 2018). Thus, we used truncated VLN3 fragments in this assay. To confirm VLN3 phosphorylation in vivo, we generated antibodies that specifically recognize pSer779 (Supplemental Fig. 3b,c). By immunoblotting with this antibody, we confirmed that full-length VLN3 is phosphorylated upon flg22 activation and MPK3/6 is required for this phosphorylation (see Fig. 2h).

5. Line 159 onwards: It is not entirely clear why the authors focussed exclusively on MPK3/MPK6. As mentioned by them, reference #22 (Berriri et al) showed that elevated MPK4 activity compromised response to sprayed bacteria (but not hand-infiltrated bacteria) and is hence involved in stomatal response. That paper also showed a strong, almost exclusive, expression of MPK4 (using pMPK4-MPK4:GFP transgenic lines) in the guard cells. There are also several other papers that report involvement of MPK4 in stomatal closure. It would be better to include MPK4 in this work, in part also to show specificity for their assays (e.g. those used in Fig 2B-E).

AU response: We included MPK4 in revised MS as suggested by the reviewer. The interaction between VLN3 and MPK4 was examined by split-LUC assays and we found no interaction between these two proteins (Fig. 2b). Moreover, we failed to detect VLN3 phosphorylation by MPK4 in the kinases assay (Fig. 2d), suggesting that VLN3 is specifically phosphorylated by MPK3/6.

6. In this context, is the phosphoshift of VLN3 in the Phos-Tag analysis lost in the mpk3mpk6 mutant? (This would lend further support the anti-pS/pT-based assay, Fig 2G).

AU response: As shown in Fig. 2g, the flg22-induced phosphoshift of C-VLN3 was lost in the *mpk3/6* mutant. Immunoblotting with α -pS779 revealed that VLN3 Ser779 was mostly unphosphorylated in nonstimulated WT but became strongly phosphorylated upon stimulation with flg22 (Fig. 2h), indicating that this residue is specifically phosphorylated in a flg22-dependent manner. In contrast, the Ser779 phosphorylation was abolished in *mpk3/6* double mutants (Fig. 2h), indicating that MPK3/6 are indeed required for the flg22-induced phosphorylation of VLN3 Ser779 in the plant cell.

Actin severing assay:

7. Figure S3: Generally, a better description of the analysis performed will help readers (not familiar with this actin filament severing assay) to interpret the data. Furthermore, in Fig S3A, the second and third lanes have different outcome although they are labelled with identical conditions (mislabelled?).

AU response: Descriptions of high-/low-speed cosedimentation assays were now included in the main text and the method section (highlighted line 279~287, line 692~711).

We thank the reviewer to point this out, this is now corrected (see Supplemental Fig. 5a).

8. In this context, shouldn't the phospho-mimetic of VLN3 have more severing activity (Fig S5C)? There seems to be no differences for the actin signals in the pellet fractions (between no VLN3, lane 1, and adding the various VLN3 variants, lanes 2-4).

AU response: The bulk actin assay might not be sensitive enough to reflect the enhanced severing activity of VLN3^{S779D} (Supplemental Fig. 6c). Thus, we performed TIRF assay and observed that VLN3^{S779D} show higher severing rate than wild-type VLN3 (Fig. 4g,h), suggesting that phospho-mimic of VLN3 have more severing activity.

Inoculation method:

9. For validating their hypothesis on stomatal defense (in Figure 8), they only tested spray inoculation to compare the VLN phospho-variants. However, Fig 1A and B also demonstrated that the *vln3* knock-out mutants are also compromised in immunity after hand-infiltration. Here, it will be prudent to additionally assess susceptibility of the complemented lines by bacteria infiltration in parallel. Possibly, a “bacterial entry” assay (as in Fig 1F) should have been tested. Such a comparison between spray and infiltrated inoculation may provide a clearer idea of the contribution of stomatal immunity versus post-invasive immunity (when the bacteria have entered the intercellular spaces of the leaves).

AU response: We performed bacterial growth assay on the complemented lines by both spray and infiltrated inoculation. The data showed that phospho-null VLN3^{S779A} failed to restore the bacterial susceptibility of *vln3* mutant to WT level in both spray-inoculation (Fig. 3b; Supplemental Fig. 4a) and hand-infiltration (Supplemental Fig. 4b) assay, suggesting that VLN3 phosphorylation by MPK3/6 is involved in both stomatal and apoplastic immunity. Given that MPK3/6 and VLN3 have broad expression patterns and MPK3/6 also function in both stomatal and apoplastic defense (Bush and Krysan, 2007; Winter et al., 2007; Bao et al., 2012; van der Honing et al., 2012; Su et al., 2017), our data suggest that MPK3/6-mediated VLN3 phosphorylation is operational in a broad range of cells.

Other points:

10. Please check the statistics tests used throughout the manuscript. For instance, Fig S1C should have been a 2-way ANOVA analysis to separate effects of genotype from treatment. In several figures, letters are used to denote statistically distinct groups but described as student's t-test (which is only pairwise test and not for multiple comparisons). Vice versa, another example is Fig 7B, which is stated as ANOVA in the figure legend but based on the description, is in fact only pair-wise comparisons (If this was indeed a Dunnett's ANOVA assay, please state it specifically as such).

AU response: Statistics tests have been checked and corrected accordingly.

11. Labelling in Fig 3B? (difference between black and orange columns?). Same issue for Fig S4B.

AU response: Corrected.

12. Fig. S2B: Composite/compiled figures should indicate spliced blots (e.g. with vertical line).

AU response: This figure is replaced with unspliced blots (see Supplemental Fig. 3a).

13. MAMP treatment labels missing in Fig S6.

AU response: Fixed.

14. Fig. 5. It would be useful to readers if they briefly described what parameters are being compared in the “correlation coefficient” analysis.

AU response: Correlation coefficient analysis provides a global view of changes in actin organization over time by calculating the correlation of the intensity of GFP-lifeactin signal at all pixel locations between all temporal pairs of images. The extent of actin rearrangements over time was reflected by decay in the correlation coefficient as the temporal interval increased. The correlation coefficient values from actin arrays with reduced rearrangement will decay less than a control population (Vidali et al., 2010). This is now included in the text (highlighted line 401~407).

In response to Reviewer #3's comments:

The manuscript by Zou et al demonstrates an axis involves MPK3/MPK6-VLN3 in flg22-induced stomatal closure and stomatal immunity against Pst DC3000. In general, I feel the work is well designed and experiments are carefully performed. I have only one major concern as follows:

As is shown in Fig 2F and Supplemental Fig 2B, HA-C-VLN3T777A, S779A is still responsive to flg22 treatment. Though the highest band is abolished upon flg22 treatment in HA-C-VLN3T777A, S779A, the lowest band was reduced and the signal for the middle band was enhanced upon flg22 treatment. Thus, other phosphorylation sites also contribute to flg22-induced band shift of C-VLN3. This notion is supported by the data in Supplemental Fig 2B, where it shows that the signal of the middle band was enhanced in HA-C-VLN3T849A, S850A upon flg22 treatment.

As is shown in Fig 2E, MPK3 and MPK6 can still induce band shift of HA-C-VLN3T777A, S779A, i.e. there are other phosphorylation sites of VLN3 by MPK3/MPK6, possibly S850. A combination of further mutations should be tested. Furthermore, if T777 and S779 are the only two phosphorylation sites by MPK3/MPK6 in response to flg22, a VLN3T777D, S779D version should constitutively regulate actin dynamics. However, this prediction was not observed in Fig 7A and B, and Fig 8A and B.

Taken together, I strongly suggest the authors further confirm the MPK3/MPK6 phosphorylation sites in VLN3 biochemically and physiologically.

AU response: We agree with the reviewer that there are additional phosphorylation sites contribute to VLN3 phosphorylation and acknowledged this in the revised text (Highlighted line 208~209). However, the role of MPK3/6-induced VLN3 phosphorylation in regulating actin dynamics during stomatal defense can be uncovered by our investigation on S779.

First, when we repeated the phosphosite mapping experiments, we noticed that, although S779A largely reduces but does not completely abolish the motility shift of C-VLN3 in some experiments; other times, however, flg22-induced phosphoshift of C-VLN3 is absent with S779A mutation (see Figure below). We reason that these differences might result from variations in the degree of immune activation, and thus the extent of Ser779 contribution may vary. Nevertheless, these results suggest that flg22-induced VLN3 phosphorylation preferentially occur at S779. In updated Fig. 2e, we repeated the in vitro kinase assays using full-length VLN3 to ensure correct protein folding and further validated the phosphorylation sites of VLN3 by MPK3/6. The phosphorylation signal was detected by [γ -³²P]ATP autoradiography and the pSer band intensity of these mutant proteins were quantified. Phosphorylation of VLN3 is greatly reduced with Ser779 single substitution (Fig. 2e). We also include S850 to test the combination of further mutations. S850 was mutated to A individually or in combination with S779 and examined their effects on VLN3 phosphorylation both in vitro and in vivo. Mutating S850 alone only shows mild or little impact on VLN3 phosphorylation. Moreover, the effect of S779/850 double mutations are not stronger than S779 single mutation (Fig. 2e, Supplemental Fig. 3a), suggesting that S779 is an important phosphosite of MPK3/6. We also generated antibodies that specifically recognize phosphorylated S779 (α -pS779).

Immunoblotting with α -pS779 revealed that VLN3 Ser779 was mostly unphosphorylated in nonstimulated WT but became strongly phosphorylated upon stimulation with flg22 (Fig. 2h), indicating that this residue is specifically phosphorylated in a flg22-dependent manner. In contrast, the Ser779 phosphorylation was abolished in *mpk3/6* double mutants (Fig. 2h). Collectively, data above demonstrate that S779 is a critical site for flg22-induced VLN3 phosphorylation, and MPK3/6 is responsible for Ser779 phosphorylation in the plant cell.

Second, the functional importance of S779 phosphorylation was further examined. In regards to actin regulation, S779 single mutation is sufficient to abolish the enhanced severing activity caused by MPK3/6-mediated VLN3 phosphorylation (Fig. 4g,h; Supplemental Fig. 7f,g), whereas VLN3 S850 mutant proteins have similar biochemical activities with WT VLN3 (Supplemental Fig.6d). We generated VLN3S779A/D, S850A/D and S779/850A/D complementary lines and examined flg22-induced stomatal closure on these lines and their susceptibility to bacteria. Importantly, lines expressing VLN3S779A have similar phenotype with *vlm3* mutant, whereas VLN3S779D is able to restore the defects of *vlm3* mutant to the WT level. Expressing VLN3S850A or VLN3S850D can fully restore the *vlm3* mutant phenotype, and the effect of S779/850 double mutations are similar with S779 single mutation (Fig. 3 and Supplemental Fig. 4). Taken together, our biochemical and physiological evidence all confirm that S779 is a key phosphosite of MPK3/6 upon flg22-induced innate immune activation. Additionally, VLN3S779 phosphorylation is biologically important for stomatal defenses. Therefore, we think we provide strong evidence to support the conclusion of this study that VLN3 phosphorylation by MPK3/6 is critical for actin dynamics during stomatal immunity.

To answer why phosphomimic version of VLN3 shows no constitutive activity, our in vitro TIRF analysis showed that VLN3S779D display an enhanced severing activity in the absence of kinases, indicating the constitutive activity in the in-vitro system (Fig. 4e,g,h). Expressing VLN3S779D fail to elicit actin response or stomatal defense in unstimulated plant, indicating that phosphorylation of S779 is required, but not sufficient, for flg22-induced actin and defense responses. This can be explained by the involvement of additional VLN3 phosphosites, as pointed out by the reviewer. Moreover, in plant cells, the activity of VLN3 might be regulated by multiple

upstream regulatory factors in addition to phosphorylation regulation. For example, immune signaling molecules such as Ca^{2+} and phospholipid may also modulate VLN3-dependent actin dynamics (Kumar et al., 2004; Khurana et al., 2010).

MAPKs are proline-dependent kinases, a proline residue always follows the phosphorylation sites. It should be careful to conclude that T777 is a phosphorylation site of MPK3/MPK6. And this should be confirmed by comparing single T777A, S779A mutations and double T777A S779A mutation.

AU response: We thank the reviewer to point this out. We compared single T777A, S779A mutations and double T777A S779A mutation. As shown below, neither phosphorylation nor function of VLN3 are affected by T777A mutation both in vitro and in vivo. It is possible that the T777 phosphorylation detected was produced by the activity of an *E. coli* kinase. All functional analysis were now performed with Ser779 alone.

Figure legend: (a) Protoplast expressing WT and mutant forms of C-VLN3 were treated with flg22. Total protein were subjected to phos-tag analyses. (b) In vitro kinase assay using activated MPK3/6 and VLN3 mutant proteins. Signals were detected by autoradiography after gel electrophoresis. (c) Stomatal aperture measurements on WT and *vl*n3 mutant carrying different VLN3 variants after mock and flg22 treatment.

Minors:

Such as Line 113, “dominate” should be “dominant”.

AU response: Corrected (highlighted line 127).

Reference

- Bao C, Wang J, Zhang R, Zhang B, Zhang H, Zhou Y, Huang S** (2012) Arabidopsis VILLIN2 and VILLIN3 act redundantly in sclerenchyma development via bundling of actin filaments. *Plant J.* **71**: 962-975
- Bi G, Zhou Z, Wang W, Li L, Rao S, Wu Y, Zhang X, Menke FLH, Chen S, Zhou J-M** (2018) Receptor-like cytoplasmic kinases directly link diverse pattern recognition receptors to the activation of mitogen-activated protein kinase cascades in *Arabidopsis*. *Plant Cell* **30**: 1543-1561
- Bush SM, Krysan PJ** (2007) Mutational evidence that the Arabidopsis MAP kinase MPK6 is involved in anther, inflorescence, and embryo development. *J. Exp. Bot.* **58**: 2181-2191
- Winter D, Ben V, Hardeep N, Ron A, Wilson GV, Provart NJ, Ivan B** (2007) An "electronic fluorescent pictograph" browser for exploring and analyzing large-scale biological data sets. *PLoS One* **2**: e718
- Guo Y, Zhao S, Jiang Y, Zhao Y, Huang S, Yuan M, Zhao Y** (2016) Casein kinase1-like protein2 regulates actin filament stability and stomatal closure via phosphorylation of actin depolymerizing factor. *Plant Cell* **28**: 1422-1439
- Jiang K, Sorefam K, Deeks MJ, Bevan MW, Hussey PJ, Hetherington AM** (2012) The ARP2/3 complex mediates guard cell actin reorganization and stomatal movement in Arabidopsis. *Plant Cell* **24**: 2031-2040
- Khurana P, Henty JL, Huang S, Staiger AM, Blanchoin L, Staiger CJ** (2010) Arabidopsis VILLIN1 and VILLIN3 have overlapping and distinct activities in bundle formation and turnover. *Plant Cell* **22**: 2727-2748
- Kumar N, Zhao PL, Tomar A, Galea CA, Khurana S** (2004) Association of villin with phosphatidylinositol 4,5-bisphosphate regulates the actin cytoskeleton. *J. Biol. Chem.* **279**: 3096-3110
- Su J, Zhang M, Zhang L, Sun T, Liu Y, Lukowitz W, Xu J, Zhang S** (2017) Regulation of stomatal immunity by interdependent functions of a pathogen-responsive MPK3/MPK6 cascade and abscisic acid. *Plant Cell* **29**: 526-542
- van der Honing HS, Kieft H, Emons AMC, Ketelaar T** (2012) Arabidopsis VILLIN2 and VILLIN3 are required for the generation of thick actin filament bundles and for directional organ growth. *Plant Physiol.* **158**: 1426-1438
- Vidali L, Burkart GM, Augustine RC, Kerdavid E, Tüzel E, Bezanilla M** (2010) Myosin XI is essential for tip growth in *Physcomitrella patens*. *Plant Cell* **22**: 1868-1882
- Zhao Y, Zhao S, Mao T, Qu X, Cao W, Zhang L, Zhang W, He L, Li S, Ren S, Zhao J, Zhu G, Huang S, Ye K, Yuan M, Guo Y** (2011) The plant-specific actin binding protein SCAB1 stabilizes actin filaments and regulates stomatal movement in *Arabidopsis*. *Plant Cell* **23**: 2314-2330
- Zheng W, Jiang Y, Wang X, Huang S, Yuan M, Guo Y** (2019) AP3M harbors actin filament binding activity that is crucial for vacuole morphology and stomatal closure in *Arabidopsis*. *PNAS* **116**: 18132-18141

REVIEWERS' COMMENTS

Reviewer #1 (Remarks to the Author):

The authors have carefully considered my comments on the previous version of the manuscript and made the required modifications accordingly.

Reviewer #2 (Remarks to the Author):

The authors have substantially revised the manuscript and demonstrated an important role of MPK3/6-mediated S779 phosphorylation of the cytoskeletal component, VLN3 villin, for regulating stomatal immunity. The findings provide mechanistic insights into the MAMP/pathogen-induced regulation of actin dynamics for stomatal closure and are therefore important for the field.

I am generally contented with the science but have to mention that the writing needs to be improved. (I seriously doubt a lay person or even someone in related field can digest the information in this current version). This entails the tricky task of explanation of a highly complex topic clearly but also concisely. Besides the need for this contextual improvement, a thorough proof-reading for typos (both text and often labels in figures) and language polishing/editing is recommended.

Comments:

I only have “minor” technical issues, mostly related to data presentation and clarity for readers:

I still have a problem with how the statistics are described/displayed. I refer to Fig 5f and Fig 7b/c to illustrate my point: In most situations, they have genotype and treatment as 2 parameters and hence a 2-way ANOVA (with post-hoc test) would be appropriate. In most cases, they “only” perform t-tests. Additionally, their usage of alphabets to mark statistical significance is not the usual convention for ANOVA. A letter would normally be used for each sample to distinguish statistical differences – including the WT (mock) sample (e.g. Fig 7b would be marked with a,b; b,b; b,b?). Here I suggest consultancy with the journal’s statistics experts on preferred presentation style.

Clarity:

Prior knowledge to certain topics are often assumed by the authors, for example, the conditional mpk3mpk6 mutant (MPK6SR; most readers will not know the purpose of the NAPP1 inhibitor without a

priori knowledge on the “Shokat” system or that the mpk3mpk6 double mutant is in fact embryo lethal if not compensated with the ATP-pocket-modified MPK6 allele).

Generally, better description of figure legends is needed. E.g. Fig 3: Are L1 and L2 are independent lines? (or what is the difference between Fig 3c and 3d?). Sometimes, it is unclear what is being shown (e.g. Fig S3d.)

L167/Fig 2: C-VLN3 is first mentioned here and should include description of which region of the protein it represents. (Same applies to the N-VLN3, Currently, this info is only found in the material methods)

Labels within Fig 2b are barely readable (despite digital zoom in).

The assay with MPK4 is not convincing. Based on the autophosphorylation profile in Fig 2d, there is little kinase activity compared to MPK3/6 (activated by MKK5-DD.), which weakens the interpretation. However, I understand that MPK4 is not their main focus. Nevertheless, I think they should at least mention this as a caveat in the interpretation. One could explain that because MPK4 did not interact in the split-LUC assay, they focussed on MPK3/6 but cannot exclude a role of MPK4. This could then be picked up in the discussion on which other kinases might play a role in the VLN3 phosphocode when considering the high expression of MPK4 in guard cells (compared to the more ubiquitous expression of MPK3/6).

Line 128-133: Revise statement: Both stomatal and basal resistance are affected! Fig 1b shows that post-invasive immunity (i.e. by hand infiltration) is also affected.

L164: Report as “no observable shift” rather than “mobility-shift was subtle”

Reviewer #3 (Remarks to the Author):

I am satisfied with this revised version. The authors answered all my concerns. This revised version is largely improved and more focus than the previous one. By generating an anti-p Ser779 antibody and stable transgenic lines, the authors provided clear and strong evidence that MPK3/MPK6-VLN3 pathway regulates actin dynamics during stomatal immunity in Arabidopsis.

In this revised version, the authors provided detailed methods for “PTI response analyses”. Line 606-607, there is no phosphatase inhibitors in their protein extraction buffer. Phosphatase inhibitors are very important when performing in vivo phosphorylation assay. Some phosphorylation events are hard to be detected or consistently repeated in the absence of phosphatase inhibitors (10 mM NaF and/or 2 mM Na₃VO₄). Phosphorylation assay of VLN3 at Srt779 in this manuscript is clear and supported by multiple evidence, especially by the newly generated anti-pSer779 VLN3 antibody. If the authors keep working on phosphorylation related research in future, please adding phosphatase inhibitors in your protein extraction buffer.

REVIEWERS' COMMENTS

In response to Reviewer #1's comments

Reviewer #1 (Remarks to the Author):

The authors have carefully considered my comments on the previous version of the manuscript and made the required modifications accordingly.

AU response: We thank this reviewer for the positive feedback on our revision.

In response to Reviewer #2's comments

Reviewer #2 (Remarks to the Author):

The authors have substantially revised the manuscript and demonstrated an important role of MPK3/6-mediated S779 phosphorylation of the cytoskeletal component, VLN3 villin, for regulating stomatal immunity. The findings provide mechanistic insights into the MAMP/pathogen-induced regulation of actin dynamics for stomatal closure and are therefore important for the field.

I am generally contented with the science but have to mention that the writing needs to be improved. (I seriously doubt a lay person or even someone in related field can digest the information in this current version). This entails the tricky task of explanation of a highly complex topic clearly but also concisely. Besides the need for this contextual improvement, a thorough proof-reading for typos (both text and often labels in figures) and language polishing/editing is recommended.

AU response: We thank this reviewer for taking the time to assess our revision. We took these advices very seriously and polished our writings as best as we can. We double-checked the text and labels in figures to make sure all typos were corrected. We believe this revision is largely improved.

Comments:

I only have "minor" technical issues, mostly related to data presentation and clarity for readers:

I still have a problem with how the statistics are described/displayed. I refer to Fig 5f and Fig 7b/c to illustrate my point: In most situations, they have genotype and treatment as 2 parameters and hence a 2-way ANOVA (with post-hoc test) would be appropriate. In most cases, they "only" perform t-tests. Additionally, their usage of alphabets to mark statistical significance is not the usual convention for ANOVA. A letter would normally be used for each sample to distinguish statistical differences – including the WT (mock) sample (e.g. Fig 7b would be marked with a,b; b,b; b,b?). Here I suggest consultancy with the journal's statistics experts on preferred presentation style.

AU response: As suggested by the reviewer, all statistic tests are now performed with two-way ANOVA (with Tukey's multiple comparisons test, or with Šídák's multiple comparisons test in Fig 3a, 5b,f). Statistical significances are indicated by different letters.

Clarity:

Prior knowledge to certain topics are often assumed by the authors, for example, the

conditional mpk3mpk6 mutant (MPK6SR; most readers will not know the purpose of the NAPP1 inhibitor without a priori knowledge on the “Shokat” system or that the mpk3mpk6 double mutant is in fact embryo lethal if not compensated with the ATP-pocket-modified MPK6 allele).

AU response: The information about the conditional *mpk3/6* double mutant was now added in the revised manuscript (Line 286).

Generally, better description of figure legends is needed. E.g. Fig 3: Are L1 and L2 are independent lines? (or what is the difference between Fig 3c and 3d?).

Sometimes, it is unclear what is being shown (e.g. Fig S3d.)

AU response: L1 and L2 in Fig 3c,d indicate two independent transgenic lines. Legends of Fig 3 and Fig S3d were modified accordingly.

L167/Fig 2: C-VLN3 is first mentioned here and should include description of which region of the protein it represents. (Same applies to the N-VLN3, Currently, this info is only found in the material methods)

Labels within Fig 2b are barely readable (despite digital zoom in).

AU response: The region of truncated VLN3 was now included in the text (Line 224). The labels in Fig 2b were now made more visible.

The assay with MPK4 is not convincing. Based on the autophosphorylation profile in Fig 2d, there is little kinase activity compared to MPK3/6 (activated by MKK5-DD.), which weakens the interpretation. However, I understand that MPK4 is not their main focus. Nevertheless, I think they should at least mention this as a caveat in the interpretation. One could explain that because MPK4 did not interact in the split-LUC assay, they focussed on MPK3/6 but cannot exclude a role of MPK4. This could then be picked up in the discussion on which other kinases might play a role in the VLN3 phosphocode when considering the high expression of MPK4 in guard cells (compared to the more ubiquitous expression of MPK3/6).

AU response: We modified our interpretation as suggested by the reviewer (Line 269) and also pointed out the potential role of MPK4 in VLN3 phosphorylation in the Discussion (Line 2007).

Line 128-133: Revise statement: Both stomatal and basal resistance are affected! Fig 1b shows that post-invasive immunity (i.e. by hand infiltration) is also affected.

AU response: Revised as suggested by this reviewer (Line 170).

L164: Report as “no observable shift” rather than “mobility-shift was subtle”

AU response: Modified (Line 221).

In response to Reviewer #3's comments

Reviewer #3 (Remarks to the Author):

I am satisfied with this revised version. The authors answered all my concerns. This revised version is largely improved and more focus than the previous one. By generating an anti-p Ser779 antibody and stable transgenic lines, the authors provided clear and strong evidence that MPK3/MPK6-VLN3 pathway regulates actin dynamics during stomatal immunity in Arabidopsis.

AU response: We thank this reviewer for the positive feedback on our revision.

In this revised version, the authors provided detailed methods for “PTI response analyses”. Line 606-607, there is no phosphatase inhibitors in their protein extraction buffer. Phosphatase inhibitors are very important when performing in vivo phosphorylation assay. Some phosphorylation events are hard to be detected or consistently repeated in the absence of phosphatase inhibitors (10 mM NaF and/or 2 mM Na₃VO₄). Phosphorylation assay of VLN3 at Srt779 in this manuscript is clear and supported by multiple evidence, especially by the newly generated anti-pSer779 VLN3 antibody. If the authors keep working on phosphorylation related research in future, please adding phosphatase inhibitors in your protein extraction buffer.

AU response: We thank this reviewer and will follow the suggestion in our future study.